# Reasoning Structure of Large Language Models

**Frédéric Berdoz** [1]   **Luca A. Lanzendörfer** [1]   **Fabian Farestam** [1]   **Roger Wattenhofer** [1]

## Abstract

Large reasoning models (LRMs) are often evaluated using metrics such as final-answer accuracy or token count. However, identical scores on these metrics can hide fundamentally different reasoning structures. To address this limitation, we introduce a scalable LRM benchmark of logic puzzles and a pipeline that converts unstructured traces into verifiable reasoning graphs of claims and dependencies. This turns reasoning into a structured, measurable object whose topology can be quantitatively analyzed. Building on this, we define a reasoning efficiency metric that quantifies how concentrated the model's logical flow is. Our analysis on open-source reasoning models shows that structural measurements separate behaviors that token count and accuracy conflate, providing a practical tool for diagnosing failure modes and comparing how reasoning scales with puzzle difficulty.

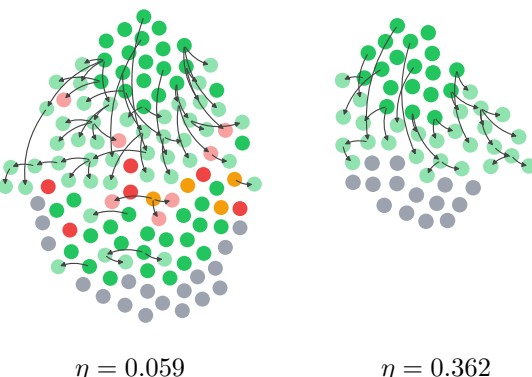

$$\eta = 0.059 \qquad \eta = 0.362$$

*Figure 1.* **Qualitative reasoning graphs.** Shown are two extracted claim graphs from two independent samples of the same model (Qwen3 235B) on the same *Tents* puzzle instance: a diffuse trace (left) and a focused trace (right). Both traces solve the instance. Nodes are atomic claims and edges indicate deductive dependencies. Node color encodes claim validity: green = verified correct, red = verified wrong, grey = unverifiable, orange = tentative. Node opacity encodes claim status: opaque = stated, semi-transparent = derived. Accuracy and token count alone are too coarse to provide meaningful feedback about how the model reasoned, whereas graph structure and efficiency (annotated $\eta$) distinguish diffuse exploration from focused deductions.

## 1. Introduction

Reasoning is central to human intelligence and remains a major challenge for machine learning systems. Thanks to their ability to exploit increased test-time compute through long Chain-of-Thought (CoT) traces (Wei et al., 2022), Large Reasoning Models (LRMs) have shown impressive performance on a broad set of reasoning tasks, including complex coding (Chen et al., 2021), logical deduction (Lin et al., 2025), mathematical reasoning (Cobbe et al., 2021), and spatial reasoning (Berdoz et al., 2026). However, because most evaluations collapse behavior into one-dimensional metrics such as final-answer accuracy or token count, it remains unclear *how* these models reason. This gap has motivated prior work to develop controllable puzzle environments that are less prone to benchmark saturation and

Code available at https://github.com/ETH-DISCO/llm-reasoning-efficiency. [1]ETH Zurich, Switzerland. Correspondence to: Frédéric Berdoz <fberdoz@ethz.ch>.

*Proceedings of the 43rd International Conference on Machine Learning*, Seoul, South Korea. PMLR 306, 2026. Copyright 2026 by the author(s).

data contamination (Chen et al., 2025a; Zhang et al., 2025). Logic puzzles have long attracted human curiosity because they are *"easy to learn, but hard to master."* Unlike many real-world tasks, they are fully specified and admit unambiguous verification. Their difficulty can be scaled without changing the underlying rules, making them an ideal controlled benchmark for studying reasoning behavior. For example, Shojaee et al. (2025) analyzed four puzzle families with adjustable complexity and found sharp regime changes and eventual collapse beyond a critical difficulty threshold. The authors also reported trace-level phenomena including overthinking, early fixation on incorrect hypotheses, and counterintuitive reductions in "reasoning effort" near failure (Shojaee et al., 2025). Together, these results suggest that controllable puzzles can reveal failure modes that accuracy alone misses, and motivate analyses that use the intermediate traces rather than only the final answer.

In this work, we move from measuring *how much* a model thinks to measuring the *structure* of its reasoning. We introduce a scalable benchmark of 2D grid puzzles and convert

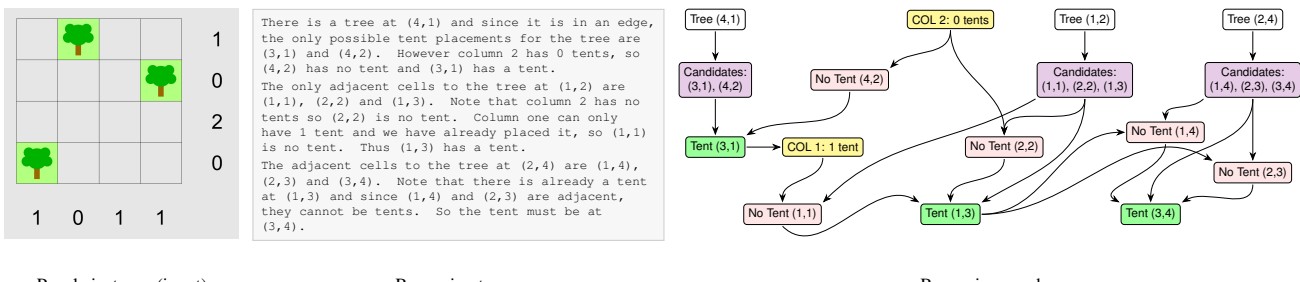

*Figure 2.* **Reasoning graph extraction.** Example of a 4×4 *Tents* instance (left), an excerpt of a *human-generated* reasoning trace (middle), and the corresponding verifiable claim graph extracted using our pipeline described in Section 3 (right). This illustrates how our pipeline turns free-form traces into structured objects that can be analyzed beyond accuracy and token count.

free-form textual traces into verifiable reasoning graphs of claims and dependencies, turning reasoning into a structured object that supports quantitative analysis. Graph topology, including depth and the fraction of the graph that supports the final solution, captures differences between focused reasoning and diffuse exploration that token count and final accuracy fail to capture. Building on this representation, we define a reasoning-flow efficiency metric $\eta$ that measures the concentration of the model's logical flow between its observations and its proposed solution. This metric can differentiate reasoning traces even when they reach the same correct solution with similar trace length. As Figure 6 shows, $\eta$ exposes differences in reasoning structure between models even in regimes where accuracy is saturated and token budgets overlap.

Analyzing reasoning traces, we find that token count is a suboptimal proxy for reasoning quality, since extra tokens primarily translate into verification overhead. In comparison, our proposed metric is able to track solution-relevant structure and correctness more effectively. Furthermore, our benchmark shows that even as models spend considerably more tokens at higher difficulty levels the hardest regime remains largely unsolved, indicating current limitations that are not resolved by simply allocating more compute.

We summarize our contributions as follows:

- We introduce a scalable benchmark of 21 logic puzzles that enables controlled scaling studies of LLM reasoning.

- We convert free-form textual reasoning traces into verifiable reasoning graphs of claims and dependencies, making reasoning topology measurable beyond accuracy and token count metrics.

- We define a reasoning efficiency metric $\eta$ that separates focused reasoning from diffuse exploration by measuring how concentrated logical flow is relative to the minimal claim set.

## 2. Related Work

### 2.1. Eliciting Reasoning in LLMs

**Prompting, search, and RL at test time.** Chain-of-Thought prompting elicits intermediate reasoning steps in language models (Wei et al., 2022). Follow-up work adds explicit test-time search, including Tree-of-Thoughts with branching and backtracking (Yao et al., 2023) and Graph-of-Thoughts (Besta et al., 2024), with adaptive variants that build task-specific DAG decompositions (Pandey et al., 2025). In parallel, reinforcement learning with verifiable rewards (RLVR) uses verifiable signals (e.g., unit tests or environment validators) to improve deliberation while reducing reward hacking concerns (Lambert et al., 2024; Amodei et al., 2016), and has enabled strong reasoning performance in recent systems (OpenAI, 2024; DeepSeek-AI, 2025). Rather than proposing new elicitation mechanisms, we analyze the structures these approaches produce by reconstructing verifiable trees from traces and studying how topology changes with puzzle complexity.

### 2.2. Benchmarking Reasoning

**Textual and mathematical reasoning.** Many benchmarks evaluate final answers on static text without an executable intermediate state. ProofWriter, ProntoQA, and LogicBench require multi-step inference but express intermediate steps in free-form language, which prevents step-level verification against an environment (Tafjord et al., 2021; Saparov & He, 2023; Parmar et al., 2024). SATBench validates final solutions with solvers, but remains text-centric rather than operating within a manipulable state machine (Wei et al., 2025). Math benchmarks such as GSM8K emphasize multi-step computation to a final output, but do not provide an environment with executable actions and verifiable state transitions (Cobbe et al., 2021).

**Controllable logical reasoning benchmarks.** As widely used benchmarks saturate, recent evaluations scale difficulty in controlled ways. GSM-Symbolic shows brittleness under

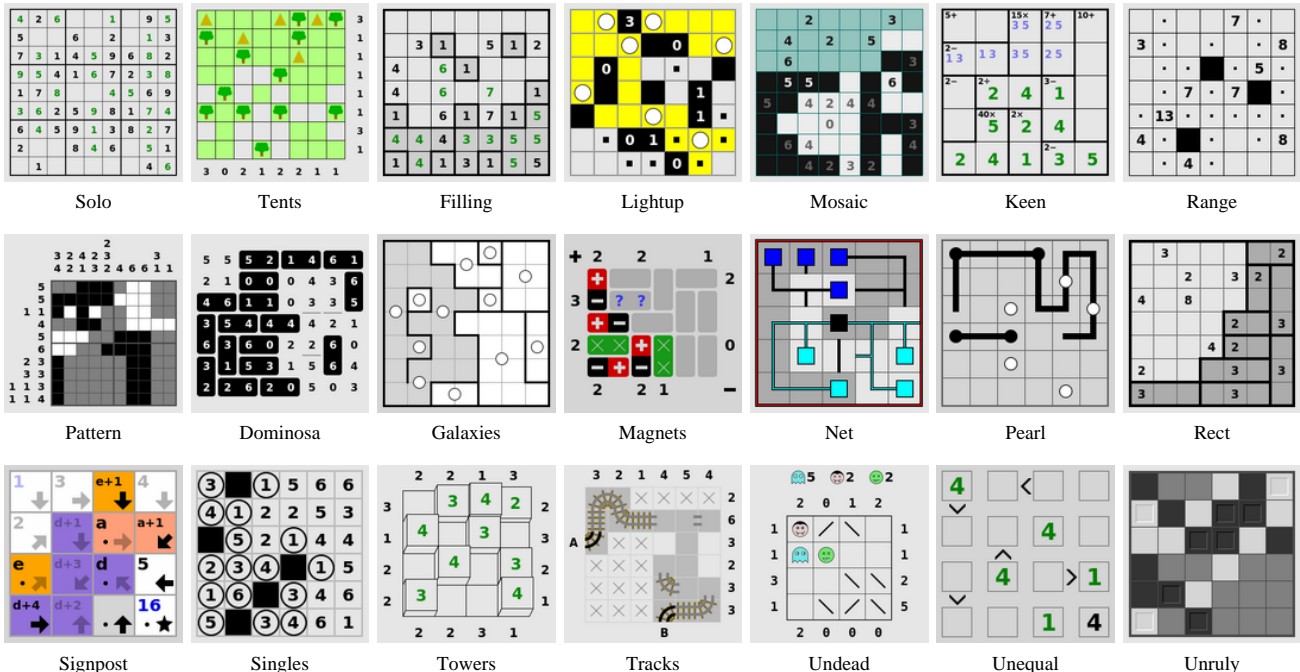

*Figure 3.* **Puzzle suite overview.** We evaluate models on 21 grid puzzles spanning diverse constraint types (e.g., placement, connectivity, counting, and Latin-square-style constraints), each with four difficulty levels (*Trivial*, *Human easy*, *Human normal*, *Human hard*). Detailed rules, state representations, and difficulty-to-size mappings for each puzzle are provided in Section E.5.

small numerical changes or irrelevant clauses (Mirzadeh et al., 2025). ZebraLogic uses scalable logic-grid puzzles and identifies a curse of complexity as constraints increase (Lin et al., 2025). SATBench generates puzzles from SAT formulas and highlights hard-UNSAT and search-related failure modes (Wei et al., 2025). While these settings typically focus on outcome metrics (e.g., accuracy, pass rates), we add structural measurements over reconstructed reasoning trees.

**Interactive games and agent evaluations.** Interactive benchmarks move toward agentic evaluation, but often lack a shared, verifiable state semantics across tasks. SmartPlay spans diverse games, which complicates defining a canonical state representation and move semantics for step-level verification (Wu et al., 2024). PuzzleBench generates dynamic visual puzzle instances with verification, but its focus is multimodal perception rather than explicit state-transition systems for search and planning (Zhang et al., 2025).

**Puzzle environments with verifiability.** Puzzle benchmarks emphasize verifiability and scalable difficulty, but differ in what is verified. Enigmata provides generator-verifier pairs for scalable evaluation and RLVR-style training, but verifies only final submissions (Chen et al., 2025a). Shojaee et al. (2025) study trace-level collapse under increasing complexity in parameterized puzzle settings, but focus on a small set of puzzle types.

### 2.3. Studying Reasoning

**Behavioral studies.** Recent work characterizes failure modes of long-trace reasoning beyond final-answer accuracy, including overthinking on trivial problems (Chen et al., 2025b), underthinking via premature switching between lines of thought (Wang et al., 2025), diversity collapse where pass@1 improves while pass@k degrades (Dang et al., 2025), and missing-premise overthinking on ill-posed questions (Fan et al., 2025). In this work, we study the structure of reasoning rather than its failure modes.

**Trace representations.** Structured trace representations support analysis and test-time selection. Landscape-style embeddings visualize convergence and provide verification signals (Zhou et al., 2026). Reasoning-graph pipelines compress traces into step graphs and link graph properties to accuracy and prompting sensitivity (Xiong et al., 2025), and other work derives graphs from hidden-state representations to study topological properties across models (Minegishi et al., 2025). Related approaches propose metrics and structures to separate useful reasoning from failed exploration, including failed-step fraction for reranking and editing (Feng et al., 2025), task-specific DAG structures and consistency metrics for math reasoning (Zhang et al., 2026), and tree-jump representations that quantify exploration and backtracking for test-time selection (Zeng et al., 2025). In contrast, our representation is grounded in an executable puzzle

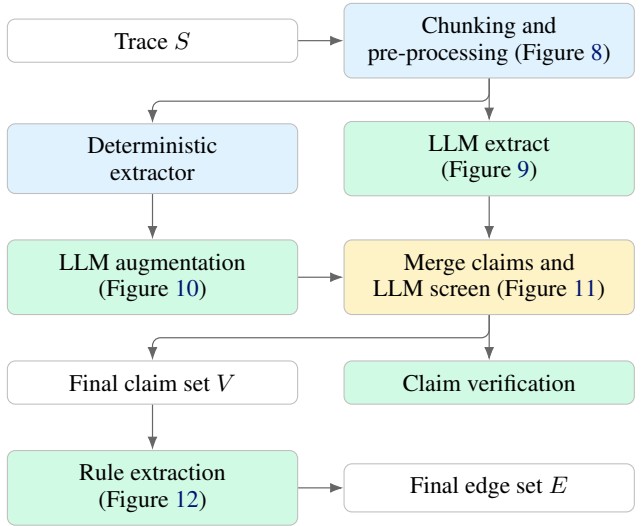

*Figure 4.* **Graph extraction overview.** Chunk-level extraction uses deterministic extraction with LLM augmentation and from-scratch LLM extraction. Outputs are merged and screened in batches. The final claim set is then processed to extract the edges that representing the canonical reasoning steps. LLM-based steps are visualized in green, pre-processing steps are shown in blue.

environment. We extract verifiable claims and dependencies: claim nodes are verified deterministically against the executable environment, while edges are attributed through constrained, puzzle-specific LLM rule application and validated by manual inspection (Section B). This grounding enables structural metrics that remain comparable across puzzle families and difficulty levels.

## 3. Methodology

We present a method for analyzing the reasoning behavior of large reasoning models (LRMs) by constructing reasoning graphs from model-generated traces. These graphs enable a unified evaluation of reasoning accuracy, structural properties, and efficiency beyond final-answer correctness and token count.

### 3.1. Solving Grid Puzzles with LRMs

We build our reasoning benchmark for LRMs on a scalable grid-based puzzle RL environment (Estermann et al., 2024) derived from Simon Tatham's puzzle collection (Tatham, 2025). These deterministic puzzles require abstraction, planning, and multi-step logical reasoning while remaining fully specified by explicit rules and state transitions. The executable environment further allows us to validate intermediate claims extracted from reasoning traces, rather than only final answers, enabling fine-grained analysis of both correctness and failure modes in the LRM's reasoning processes. Details about the prompting strategy to solve the puzzles

are provided in Section E.1.

### 3.2. From Textual Traces to Reasoning Graphs

Our goal is to assess the quality of LLM reasoning beyond downstream accuracy or token-level metrics. To this end, we extract the latent reasoning structure underlying the textual trace produced by a reasoning model on a given puzzle instance. We represent this reasoning process as a directed graph, where nodes correspond to claims and edges capture deductive dependencies. In this section, we formalize the notation and structural assumptions used to define these graphs.

**Atomic Claims.** Let $S = (s_1, s_2, \dots)$ denote the reasoning trace of a LRM on a puzzle instance, represented as an ordered sequence of sentences appearing in the generated output, and let $\mathcal{C}$ be the set of all atomic claims that can be made about a puzzle. Examples of such claims could be: "The cell at row 3 and column 4 is 9" for *Sudoku* (whose correctness only depends on the full solution), or "There is already a 4 in column 5" (whose correctness depends on the current partial solution). Note that simply stating a solution for a $9 \times 9$ *Sudoku* requires 81 claims and that a single sentence may contain multiple distinct claims due to the free form nature of text.

**Reasoning Graphs.** Let $G = (V, E)$ be a directed graph corresponding to the reasoning trace of a LRM, where $V \subseteq \mathcal{C} \times \mathbb{N}$ denotes the set of claim occurrences extracted from $S$ (each vertex $v = (c, i) \in V$ indicates that claim $c$ is asserted in sentence $s_i$). By construction, each element in $V$ is unique: the same claim cannot appear twice in the same sentence. Edges in $G$ represent either an inference from premises to a conclusion, or a restatement link between two occurrences of the same underlying claim. When an inference requires multiple premises, separate edges connect each premise to the derived claim. All premises of an inference appear before its conclusion in $S$, which implies a topological order over $V$. As a consequence, $G$ and all its subgraphs are directed acyclic graphs (DAGs). We also merge restated static claims into the first occurrence for graph metrics.

**Minimal Claim Set.** For a given puzzle instance, let $C^* \subseteq \mathcal{C}$ denote the minimal set of claims that fully determine the puzzle solution. Formally, each $c \in C^*$ is a claim that is required to reconstruct the known solution and $C^*$ is minimal in the sense that removing any $c \in C^*$ would make it impossible to recover the full solution from the remaining claims without additional inference steps. In other words, $C^*$ is simply the formatted statement of the solution and we have that $C^* \subseteq \{c \in \mathcal{C} \mid \exists (c, s) \in V \text{ for at least one } s \in S\}$ if the

---

The following are the labels within the flowchart (Figure 4):

- Trace $S$
- Chunking and pre-processing (Figure 8)
- Deterministic extractor
- LLM extract (Figure 9)
- LLM augmentation (Figure 10)
- Merge claims and LLM screen (Figure 11)
- Final claim set $V$
- Claim verification
- Rule extraction (Figure 12)
- Final edge set $E$

*Table 1.* **PUZZLE benchmark results by difficulty.** For each difficulty level we report accuracy (Acc., %) and mean completion token count (Tok.). The final columns report mean accuracy and mean token count averaged across the four difficulty levels (Avg Acc., Avg Tok.).

| Model | Trivial | | Human easy | | Human normal | | Human hard | | Avg | |
|---|---|---|---|---|---|---|---|---|---|---|
| | Acc. | Tok. | Acc. | Tok. | Acc. | Tok. | Acc. | Tok. | Acc. | Tok. |
| **GPT-5** | 83.8 | 4153.5 | 69.5 | 10179.6 | 58.1 | 17273.6 | 5.7 | 19861.9 | 54.3 | 12867.2 |
| **Qwen 3 235B** | 69.5 | 10257.3 | 44.8 | 19033.0 | 21.0 | 23104.0 | 0.0 | 23608.6 | 33.8 | 19000.7 |
| **DeepSeek V3.2** | 77.1 | 7694.7 | 53.3 | 20632.6 | 44.8 | 27037.7 | 0.0 | 36787.4 | 43.8 | 23038.1 |
| **Kimi K2** | 77.1 | 10601.5 | 56.2 | 29713.7 | 41.0 | 43751.2 | 1.0 | 61307.1 | 43.8 | 36343.4 |

LRM solves the puzzle. The converse is not true however, since the LRM might give a correct solution somewhere in its reasoning trace but submit a different solution. We define $V_{\text{sol}}^* = \{(c, s_i) \in V \mid c \in C^*, i = \min\{j \mid (c, s_j) \in V\}\}$ as the set of first occurrences of claims that belong to the solution.

**Reasoning Subgraphs for Solved Puzzles.** We further identify two subgraphs for solved puzzle instances. First, we define the minimal solution subgraph $G_{\text{sol}} = (V_{\text{sol}}, E_{\text{sol}})$ of $G$ as the subgraph containing all vertices that directly or indirectly contribute to $V_{\text{sol}}^*$, formally defined as follows:

$$V_{\text{sol}} = V_{\text{sol}}^* \cup \{ v \in V \mid \exists u \in V_{\text{sol}}^* : v \rightsquigarrow_G u \},$$

$$E_{\text{sol}} = \{(u, v) \in E \mid u, v \in V_{\text{sol}}\}.$$

We use the notation $u \rightsquigarrow_G v$ to indicate that there exists a directed path from $u$ to $v$ in $G$. Similarly, let $G_{\text{ver}} = (V_{\text{ver}}, E_{\text{ver}})$ be the verification subgraph, with

$$V_{\text{ver}}' = V_{\text{sol}}^* \cup \{ v \in V \setminus V_{\text{sol}} \mid \exists u \in V_{\text{sol}}^* : u \rightsquigarrow_G v \}$$

$$V_{\text{ver}} = V_{\text{ver}}' \cup \{ v \in V \setminus V_{\text{sol}} \mid \exists u \in V_{\text{ver}}' : v \rightsquigarrow_G u \}$$

$$E_{\text{ver}} = \{(u, v) \in E \mid u, v \in V_{\text{ver}}\}$$

Intuitively, $V_{\text{ver}}$ contains all descendants of $V_{\text{sol}}^*$ and all their ancestors that are not in $V_{\text{sol}}$.

### 3.3. Reasoning Metrics

**Graph-based Reasoning Metrics.** We report graph-size and graph-topology statistics on the extracted claim graph $G = (V, E)$ and its subgraphs $G_{\text{sol}}$ and $G_{\text{ver}}$. We use $|V|$, $|V_{\text{sol}}|$, and $|V_{\text{ver}}|$ to denote the number of claim nodes in each graph. We define the depth of a node $v$ as the length of the longest directed path from any stated node to $v$, and the depth of the graph as the maximum depth of all nodes. For solved instances, we quantify (i) how much of the extracted graph supports the minimal solution subgraph ($|V_{\text{sol}}|/|V|$), and (ii) how much additional reasoning is devoted to verification ($|V_{\text{ver}}|/|V_{\text{sol}}|$).

**Validity-based Reasoning Metrics.** Each extracted verifiable claim is checked against the executable puzzle environment (see Section B), yielding a correctness label for each claim. To localize drift, we define *first-error depth* as the depth (in $G$) of the earliest incorrect claim.

**Flow-based Reasoning Metrics.** We aim to quantify the structure of the *logical flow* induced by the LRM's reasoning. To do so, we model its reasoning process as an *absorbing Markov chain* (Kemeny & Snell, 1960) on a directed acyclic graph (DAG), where claims extracted from the reasoning trace are transient states. Let $d^-(v)$ and $d^+(v)$ denote the in- and out-degrees of a node $v \in V$, respectively, and let $G_{\text{abs}} = (V \cup \{a\}, E \cup \{(v, a) \mid v \in V, d^+(v) = 0\})$ denote the augmented graph obtained by connecting every leaf node in $G$ to a single absorbing state $a \notin V$, into which all logical flow converges. The structure of the flow is characterized by the transition matrix $P$, i.e., the row-normalized adjacency matrix of $G_{\text{abs}}$, whose canonical form can be partitioned as follows (assuming $a$ comes after $v_{|V|}$ in the topological order):

$$P = \begin{pmatrix} Q \in \mathbb{R}^{|V| \times |V|} & R \in \mathbb{R}^{|V| \times 1} \\ \mathbf{0} \in \mathbb{R}^{1 \times |V|} & 1 \end{pmatrix},$$

with $Q$ and $R$ the transient and absorbing components of the Markov chain, respectively. We model the initial distribution of the LRM's *logical mass* on a puzzle instance as a uniform distribution over the source nodes $V_{\text{SRC}} = \{v \in V \mid d^-(v) = 0\}$. Formally, we define the initial logical mass as a row vector $\boldsymbol{\pi}$ lying on the $|V|$-dimensional simplex, where $\pi(v) = 1/|V_{\text{SRC}}|$ for $v \in V_{\text{SRC}}$ and $\pi(v) = 0$ for $v \in V \setminus V_{\text{SRC}}$. Under standard Markov chain dynamics, the distribution $\boldsymbol{\pi}$ evolves through the transient nodes according to $\boldsymbol{\pi} Q^t$, with $t \geq 0$ the discrete time step. The amount of logical mass flowing through each transient node can therefore be quantified as $\mathbf{m} = \boldsymbol{\pi} N$, with $N = \sum_{t=0}^{\infty} Q^t$ the fundamental matrix of $G_{\text{abs}}$. Taking inspiration from structural information theory (Li & Pan, 2016; Shannon, 1948), we quantify the structural entropy of the LRM's logical flow as

$$H_{\text{str}}(G) = - \sum_{v \in V} \frac{m(v)}{\|\mathbf{m}\|} \log \left( \frac{m(v)}{\|\mathbf{m}\|} \right), \quad (1)$$

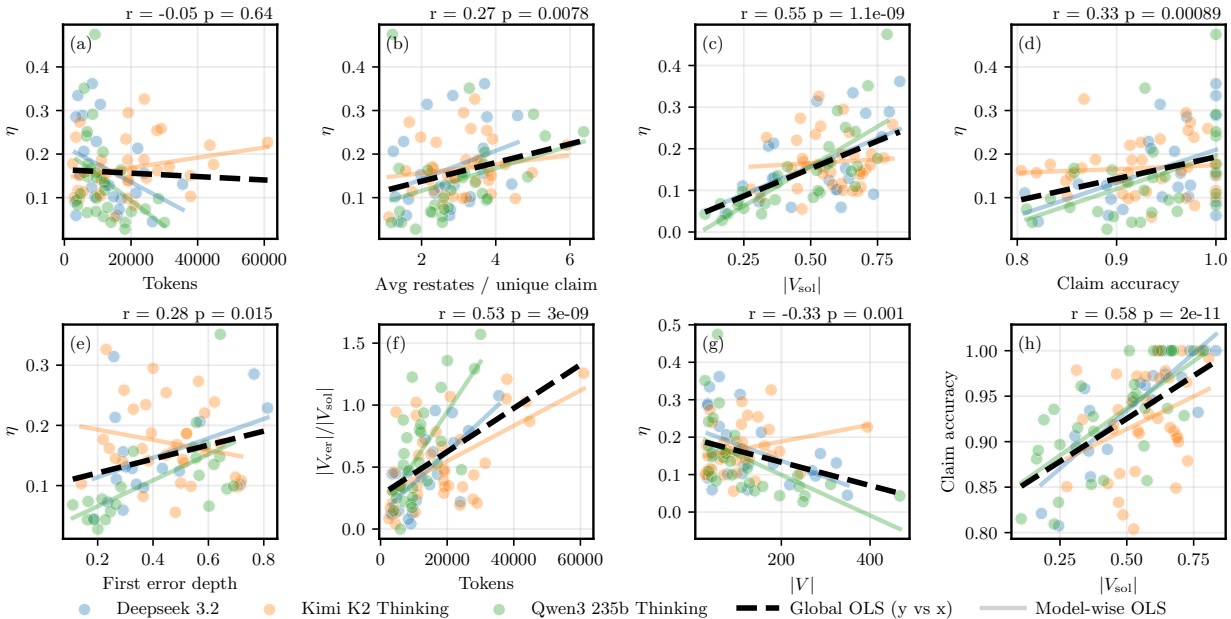

*Figure 5.* **Efficiency correlations.** Each panel plots a graph-level metric against efficiency $\eta$ or token count. Points are colored by model; dashed lines show linear fits. Statistical significance is reported above each plot. (a) Efficiency is not significantly correlated with verbosity. (b,c) Efficiency tracks claim composition: negatively with stated fraction and positively with restated fraction. This suggests that LRM reason in a more strucured manner when they restated claims rather than stating new ones. (d) Higher efficiency is associated with higher claim accuracy. (e) Higher efficiency is associated with later first-error depth. (f) Verification overhead ($|V_{\text{ver}}|/|V_{\text{sol}}|$) grows with token count.

with $\|\mathbf{m}\| = \sum_{v \in V} m(v)$. The intuition behind $H_{\text{str}}$ lies in its ability to measure the dispersion versus focus of the LRM's reasoning process across the logical graph. By treating the normalized logical mass as a probability distribution, the entropy quantifies whether the model's attention is concentrated on a single, deterministic path (low entropy) or scattered across many competing claims and inferences (high entropy). It also penalizes restated and unused claims, since both diffuse the logical flow.

**Reasoning Flow Efficiency.** We aim to evaluate the LRM's ability to minimize "structural noise" during its reasoning. To that end, given a puzzle instance, its solution $C^*$ and the LRM reasoning graph $G$, we define the *reasoning flow efficiency*

$$\eta = \frac{\log |V| - H_{\text{str}}(G)}{\log |V| - \log |C^*|}. \tag{2}$$

Intuitively, $\eta$ measures how concentrated the model's logical flow is relative to the minimal claim set that specifies the solution. For solved instances, $\eta$ is designed to lie in $[0, 1]$, with $\eta \approx 1$ for highly focused reasoning (low structural entropy relative to the gap between $\log |V|$ and $\log |C^*|$) and smaller values for more diffuse reasoning that spreads flow across many peripheral claims.

### 3.4. Reasoning Graph Construction

We now describe how reasoning graphs are extracted from model-generated textual traces. The procedure is fully automated, puzzle-agnostic at the graph level, and modular with respect to claim definitions, inference rules, and verification logic. The construction proceeds in three stages: claim extraction, rule extraction, and claim verification (see Figure 4 for an overview). The extraction pipeline is designed to maximize claim recall while conservatively attributing inference structure, ensuring that all extracted elements are grounded in the text.

**Claim Extraction.** We extract claims using a two-pass hybrid procedure that combines deterministic pattern matching with LLM-based extraction. This design balances precision and recall while reducing the reliance on a single extractor. The reasoning trace is first segmented into sentences and processed in token-balanced chunks to satisfy context-length constraints. For each chunk, we generate candidate claims using two complementary mechanisms: (i) a high-precision deterministic extractor followed by an LLM-based completion step that repairs schema violations and adds directly implied missing fields, and (ii) a high-recall LLM-based extractor operating without rules. All extracted claims are treated as event-level occurrences and retain references to their source sentences. Exact-duplicate candidates are re-

moved locally, after which claims are verified in batches of 200 by an LLM with access to a localized support window around their source text. Unsupported or ill-formed claims are discarded, and conservative normalization is applied without collapsing repeated events. Finally, claims are globally deduplicated, ordered by trace position, and assigned identifiers. The prompts used to extract the claims are included in Section E.2, while the puzzle-specific claim types are detailed in Section E.4.

**Rule Extraction.** For each finalized, non-tentative claim, we attempt to recover a single explicit rule application that explains how the claim follows from earlier claims in the trace. Rule extraction is performed once per distinct claim content to avoid redundant derivations. Claims are processed in trace order. For each claim, we construct a truncated prefix of the reasoning trace ending at its last supporting sentence, together with all previously extracted and formatted claims. Given this context, an LLM is then prompted to either (i) return a single rule application linking the claim to prior claims, or (ii) return no rule, in which case the claim is treated as directly stated. When a derivation is identified but required premises are missing from the trace, an explicit placeholder claim is inserted to mark the gap. If the same claim has already been stated one, it is classified as restated and a direct edge link the last occurrence of that claim to the restated one. The prompts used to extract the rules are provided in Section E.3, while the puzzle-specific rule types are detailed in Section E.4.

**Claim Verification.** Each extracted verifiable claim is independently verified against the executable puzzle environment. For each claim, the environment deterministically checks whether the claim is consistent with the puzzle rules, constraints and final solution. Verification results are attached to claim nodes. See Section B for implementation details.

# 4. Experiments

**Overview.** We evaluate large reasoning models on a scalable suite of deterministic grid puzzles and report both outcome metrics and structural metrics extracted from the intermediate traces. Our main benchmark results are summarized in Table 1, which reports accuracy and token usage across difficulty levels. We additionally visualize efficiency scaling trends in Figure 6, illustrate qualitative differences between diffuse and focused reasoning graphs in Figure 1, and summarize correlations between graph-based efficiency and other trace statistics in Figure 5.

## 4.1. Benchmark

We evaluate models on a fixed suite of 21 puzzles (overview in Figure 3) with four difficulty levels (*Trivial*, *Human easy*, *Human normal*, *Human hard*). Difficulty is controlled by puzzle-specific generators and corresponds to grid sizes and clue densities that vary by puzzle family (see Section E.5). For each puzzle and difficulty level, we evaluate on a fixed set of 5 instances and reuse the same instance IDs across all models to support fair comparisons. All models are prompted with the same solver prompt (see Section E.1) and must output a final solution in a standardized format. We decode using temperature sampling with $T = 1$. Because our analysis targets the reasoning *structure* rather than the peak capability, the framework is decoding-agnostic. We fix $T = 1$ for consistency, noting that very low temperatures can induce repetitive loops in reasoning models (Pipis et al., 2025) and that decoding strategy can shift model rankings (Song et al., 2025). Outputs are parsed into structured actions/placements. Malformed outputs are treated as incorrect, and final solutions are verified by the executable puzzle environment (Section B). We also report completion token usage as Tokens. We evaluate frontier and open reasoning models, including GPT-5 (OpenAI, 2025) and open models such as Qwen 3 235B (Yang et al., 2025), DeepSeek V3.2 (DeepSeek-AI et al., 2025), and Kimi K2 (Team et al., 2025). We report accuracy and token usage by difficulty in Table 1. Accuracy is computed over all runs. To complement the table, Figure 6 summarizes how efficiency changes with puzzle size across models. We provide per-puzzle breakdowns in Section D.

## 4.2. Reasoning Graph Statistics

We analyze the structure of intermediate traces by extracting claim graphs as described in Section 3.4. We use GPT-5.2 for claim extraction and screening, and GPT-5-mini for rule extraction (OpenAI, 2025). We extract graphs only from open-source reasoning models, since closed-source models usually do not provide access to their traces. We report a stability analysis of the extraction pipeline in Section A.

## 4.3. Reasoning Efficiency

**Efficiency from claim graphs.** We compute reasoning-flow efficiency $\eta$ from the extracted claim graphs on solved puzzle instances. Intuitively, higher $\eta$ indicates more concentrated logical flow relative to the minimal claim set, while lower $\eta$ indicates more diffuse exploration and verification overhead. Figure 5 summarizes how $\eta$ relates to verbosity (Tok.), claim composition, claim-level correctness, the depth at which the first incorrect claim occurs, and verification overhead.

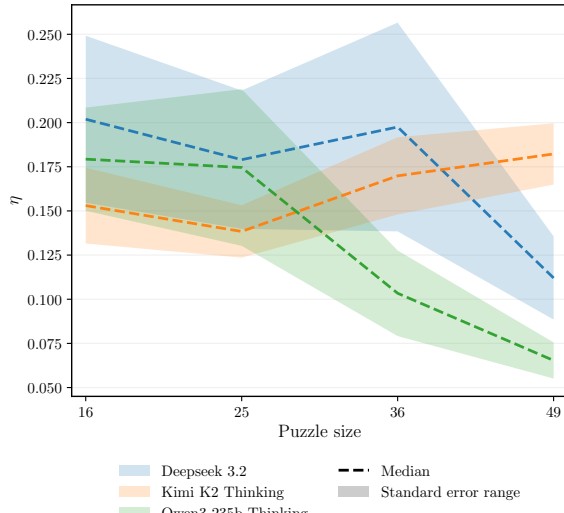

*Figure 6.* **Reasoning-flow efficiency vs. puzzle size under perfect accuracy.** Mean reasoning-flow efficiency $\eta$ as a function of puzzle size for settings where all puzzle sizes are solved. We observe that even when correctness is saturated, $\eta$ exposes differences in reasoning structure and scaling behavior, demonstrating that accuracy alone hides substantial variation in how solutions are produced.

## 5. Discussion

**Difficulty scaling exposes steep accuracy drop despite increased compute.** Across all models, accuracy declines sharply as difficulty increases from *Trivial* to *Human hard* (e.g., GPT-5 drops from 83.8% to 5.7%; Qwen3 235B drops from 69.5% to 0%; DeepSeek V3.2 drops from 76.2% to 0%; Kimi K2 drops from 77.1% to 0.95%), while mean completion tokens rise substantially (from roughly 4-11k to approximately 20-61k). Notably, more tokens do not imply better performance. Kimi K2 consistently uses the largest token budgets yet does not outperform GPT-5, whereas GPT-5 achieves the best accuracy at every difficulty while remaining the most token-efficient. DeepSeek V3.2 is competitive at *Human normal* (51.4%) but collapses at *Human hard* despite increased tokens, and Qwen3 235B degrades earlier (only 21.0% at *Human normal*) and reaches 0% on *Human hard*. Overall, the hardest regime remains largely unsolved for all models even with large token budgets, suggesting fundamental limitations in scaling reasoning beyond simply allocating more computation.

**Token count is not a proxy for reasoning quality.** We find that token count alone is a poor proxy for reasoning quality. Across runs, reasoning-flow efficiency $\eta$ is essentially uncorrelated with token count ($r = -0.05, p = 0.64$). This suggests that verbosity cannot be interpreted as either better or worse reasoning structure. Bootstrap 95% confidence intervals for all $\eta$ correlations are reported in Section C.

*Table 2.* $\eta$ **versus simpler structural metrics.** Pooled Pearson correlations with claim accuracy and with $\eta$, computed on the same graphs.

| Metric | vs. Accuracy | vs. $\eta$ |
|---|---|---|
| Depth | $-0.263$ | $+0.046$ |
| Diameter | $-0.329$ | $+0.010$ |
| Avg. path length | $-0.182$ | $+0.051$ |
| Width | $-0.618$ | $-0.431$ |
| $|V|$ | $-0.666$ | $-0.419$ |
| Tokens | $-0.576$ | $-0.120$ |
| $\eta$ | $+0.368$ | — |

$\eta$ **versus simpler structural metrics.** We test whether $\eta$ adds anything beyond elementary graph statistics (Table 2). Width, $|V|$, and token count all correlate *negatively* with accuracy and strongly with one another, indicating that they primarily track puzzle difficulty: harder instances yield larger, wider graphs with more tokens. $\eta$ is the only metric that correlates *positively* with accuracy while remaining uncorrelated with token count, because its size normalization (Equation (2)) factors out graph scale. Per-model breakdowns are consistent (Table 9).

**Extra tokens mainly translate into verification overhead.** Longer traces do not reliably improve $\eta$. These longer traces primarily manifest as increased verification overhead. In particular, token count correlates strongly with the ratio $|V_{\text{ver}}|/|V_{\text{sol}}|$ ($r = 0.53, p = 3 \cdot 10^{-9}$), indicating that additional tokens tend to be spent on checking and auxiliary reasoning around the solution rather than on expanding the core solution-supporting chain.

**Efficiency reflects on-solution focus and penalizes graph bloat.** In contrast, $\eta$ is closely linked to the amount the reasoning graph contributes to the solution. Efficiency increases considerably with the solution-supporting portion of the graph ($r = 0.55, p = 1.1 \cdot 10^{-9}$), while it decreases as the overall graph grows ($r = -0.33, p = 0.001$). This pattern is consistent with "graph bloat" from branching, detours, and redundant structure from diffuse reasoning flow.

**Correctness helps, but structure matters beyond accuracy.** Correctness is beneficial, but not sufficient. Claim accuracy shows only a moderate association with $\eta$ ($r = 0.33, p = 8.9 \cdot 10^{-4}$), suggesting that models can be largely correct but still inefficient if they accumulate many correct claims that are not necessary.

**Early errors are associated with inefficient traces.** Reasoning traces where the first incorrect claim appears later tend to be more efficient ($r = 0.28, p = 0.015$), aligning with the intuition that early errors induce drift and trigger corrective exploration that lowers reasoning efficiency.

**Some redundancy may be beneficial.** We find that not all redundancy is detrimental. We observe a positive correlation between $\eta$ and the average number of restatements per unique claim ($r = 0.27, p = 0.0078$). This may reflect beneficial "anchoring" behavior in the form of structured recap of key constraints rather than aimless repetition.

**Accuracy and on-solution reasoning reinforce each other.** Claim accuracy correlates strongly with the solution-supporting fraction of the graph ($r = 0.58, p = 2 \cdot 10^{-11}$), implying that off-solution exploration is a major source of errors. While these trends hold globally, model-wise fits differ in slope and intercept across several plots, highlighting that different models allocate tokens differently (solving vs. verifying vs. meandering). This finding reinforces that raw token counts are not directly comparable as a measure of reasoning quality.

**The signal extends below the solved subset.** While $\eta$ is defined on solved instances, a comparison against failed *Tents* traces (Section C) shows the same structure-quality relationship beyond final-answer correctness: failed traces are substantially less efficient ($\eta$ lower by more than half) and spread reasoning across larger, more diffuse graphs, with the first error emerging earlier. This indicates that $\eta$ reflects reasoning quality broadly rather than only separating solved from unsolved cases.

## 6. Limitations

A limitation of our approach is that claim and graph extraction uses LLM components and is therefore not fully static. We reduce bias by separating roles: GPT-5.2 performs semantically difficult claim extraction, while GPT-5-mini handles model-agnostic rule extraction independent of the evaluated systems. As shown in Section A, variance is modest and traces with little explicit reasoning yield no meaningful graph, a failure mode that our metrics explicitly capture.

We validate this pipeline empirically: a six-extractor ablation (Section A) yields consistent directional trends with no self-bias ($\eta$ varies by only 1.9% across extractors), manual inspection of 200 rule applications finds 75.5% fully correct under a strict criterion (a single missing premise counts as a full error), and $\eta$ is robust to graph perturbations (single-perturbation CV below 5% for $6 \times 6$ and larger graphs). A second limitation is that each puzzle family requires domain-specific claim and rule types. This is inherent to any process-level evaluation of reasoning traces. Process reward models likewise need per-task verifiers, whereas our structural layer (graph construction, Markov chain, $\eta$) is fully puzzle-agnostic.

## 7. Conclusion

This work proposes a scalable benchmark of deterministic 2D grid puzzles together with a pipeline that converts free-form reasoning traces into verifiable reasoning graphs of atomic claims and dependencies. Our proposed efficiency metric $\eta$ captures how concentrated a model's logical flow is relative to the minimal claim set defining a solution. We demonstrate this framework on *Tents*, where graph structure and $\eta$ distinguish focused deductions from diffuse exploration that accuracy and token count cannot easily capture.

Looking ahead, the structural layer (graph construction, the Markov chain, and $\eta$) is domain-agnostic; only the claim verification is puzzle-specific. The framework therefore extends to any setting where intermediate steps can be checked: mathematical reasoning, where intermediate equations admit symbolic verification, and code generation, where unit tests provide partial claim verification. Agentic tool-use settings, in which inefficient exploration is common, are a further natural target. If extraction becomes sufficiently reliable and low-latency, graph-based signals such as $\eta$ could also serve as auxiliary feedback in post-training with verifiable environments, e.g., a shaping reward favoring solution-focused reasoning while preserving correctness.

## Impact Statement

This work enables more diagnostic evaluation of LLM reasoning by converting free-form puzzle traces into verifiable reasoning graphs grounded in an executable environment and introducing a structure-sensitive metric that distinguishes solving from meandering or excessive verification beyond accuracy and token count. It can improve reproducibility, help identify early-error cascades and inefficient behaviors, and guide development toward more reliable and cost-effective reasoning. Risks include metric gaming, overfitting research to puzzle-style reasoning, potential bias from imperfect LLM-based extraction, and added compute/latency costs. These are mitigated by treating metrics as diagnostic (not a single leaderboard score) and reporting multiple additional metrics.

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

**Appendix overview.** This appendix contains (i) stability checks for the graph-extraction pipeline in Section A, (ii) claim verification details in Section B, (iii) per-puzzle performance breakdowns in Section D, (iv) all prompts used in the pipeline in Section E, and (v) puzzle rules and difficulty specifications in Section E.5.

## A. Graph Extraction Stability

Our pipeline uses LLM-based components, so the extracted graph is not guaranteed to be identical across runs. To quantify extractor variance, we repeat graph extraction three times on the *same* reasoning trace. We report stability of the unique extracted claim set $\mathcal{C}_{\text{unique}}$ (deduplicated by exact payload matching) via the pairwise Jaccard overlap between runs, and we report how this variability propagates to our downstream structural metrics ($H_{\text{str}}$ and $\eta$). Overall, repeated extraction from the same trace yields high claim-set overlap, and $H_{\text{str}}$ and $\eta$ remain comparatively stable (Table 3), supporting their use as coarse, graph-level descriptors rather than brittle artifacts of extraction.

*Table 3.* **Graph extraction stability (same-trace repeats).** We report stability under repeated extraction from the same reasoning trace (three runs). $|\mathcal{C}_{\text{unique}}|$ is the mean number of unique claims (deduplicated by exact payload matching). Jaccard is the mean $\pm$ standard deviation of pairwise Jaccard overlaps between claim sets. $H_{\text{str}}$ and $\eta$ are reported as mean $\pm$ standard deviation over runs.

| Model | $|\mathcal{C}_{\text{unique}}|$ | Jaccard | $H_{\text{str}}$ | $\eta$ |
|---|---|---|---|---|
| Kimi-K2 Thinking | 22.7 | $0.89 \pm 0.08$ | $3.184 \pm 0.191$ | $0.946 \pm 0.059$ |
| DeepSeek V3.2 | 25.0 | $0.79 \pm 0.07$ | $3.573 \pm 0.092$ | $0.840 \pm 0.022$ |
| Qwen3 235B 2507 Thinking | 37.7 | $0.98 \pm 0.01$ | $3.660 \pm 0.063$ | $0.820 \pm 0.014$ |

### A.1. Robustness to Extractor Choice

To verify that our structural findings are not an artifact of using GPT-5.2 as the claim extractor, we repeat extraction with five additional LLMs on the same reasoning traces. Directional trends are consistent across all six extractors: the DeepSeek source yields the lowest $\eta$ for every extractor (6/6), and no extractor systematically inflates $\eta$ on traces from its own model family (no self-bias). Across extractors, $\eta$ has a coefficient of variation of 1.9%.

*Table 4.* **Robustness to extractor choice.** Graph-extraction statistics when the claim extractor is varied across six LLMs (rows), applied to traces from three source models. Values are mean $\pm$ std over three runs. *Deterministically extracted solution claims are excluded*, so $\eta$ here isolates model-generated reasoning and is not directly comparable to Table 3; Jaccard values are conservative lower bounds.

| Extractor | Source | Jaccard | $H_{\text{str}}$ | $\eta$ |
|---|---|---|---|---|
| Main (GPT-5.2) | Kimi | $0.89 \pm 0.08$ | $3.38 \pm 0.09$ | $0.15 \pm 0.06$ |
| | DeepSeek | $0.79 \pm 0.07$ | $3.39 \pm 0.07$ | $0.14 \pm 0.06$ |
| | Qwen3 | $0.98 \pm 0.01$ | $3.66 \pm 0.06$ | $0.15 \pm 0.04$ |
| GPT-5.4 | Kimi | $0.81 \pm 0.07$ | $3.47 \pm 0.25$ | $0.21 \pm 0.08$ |
| | DeepSeek | $0.98 \pm 0.01$ | $3.40 \pm 0.06$ | $0.13 \pm 0.06$ |
| | Qwen3 | $0.92 \pm 0.04$ | $3.43 \pm 0.22$ | $0.21 \pm 0.05$ |
| DeepSeek | Kimi | $0.88 \pm 0.06$ | $3.45 \pm 0.22$ | $0.20 \pm 0.08$ |
| | DeepSeek | $0.98 \pm 0.02$ | $3.40 \pm 0.07$ | $0.11 \pm 0.04$ |
| | Qwen3 | $0.91 \pm 0.01$ | $3.44 \pm 0.20$ | $0.21 \pm 0.05$ |
| Qwen3 | Kimi | $0.92 \pm 0.03$ | $3.32 \pm 0.03$ | $0.20 \pm 0.10$ |
| | DeepSeek | $0.90 \pm 0.03$ | $3.41 \pm 0.08$ | $0.10 \pm 0.03$ |
| | Qwen3 | $0.92 \pm 0.01$ | $3.31 \pm 0.03$ | $0.29 \pm 0.05$ |
| Kimi | Kimi | $0.67 \pm 0.06$ | $3.42 \pm 0.22$ | $0.22 \pm 0.10$ |
| | DeepSeek | $0.90 \pm 0.07$ | $3.37 \pm 0.05$ | $0.12 \pm 0.03$ |
| | Qwen3 | $0.71 \pm 0.09$ | $3.46 \pm 0.22$ | $0.20 \pm 0.05$ |
| GPT-OSS | Kimi | $0.83 \pm 0.07$ | $3.40 \pm 0.21$ | $0.22 \pm 0.09$ |
| | DeepSeek | $0.65 \pm 0.05$ | $3.38 \pm 0.20$ | $0.21 \pm 0.09$ |
| | Qwen3 | $0.77 \pm 0.10$ | $3.33 \pm 0.09$ | $0.22 \pm 0.04$ |

### A.2. Sensitivity to Markov-Chain Assumptions

Our flow metric assumes a uniform initial distribution $\pi$ over source nodes and unweighted (row-normalized) transitions $P$. We test seven alternatives against this default. All transition-weighting variants preserve near-perfect instance ranking;

initial-distribution variants shift more but remain strongly rank-correlated. The variation in $\eta$ stems primarily from the normalization rather than the underlying entropy, confirming that $\eta$ captures intrinsic graph structure.

*Table 5.* **Sensitivity to Markov-chain modeling choices.** Each row is an alternative to the default (uniform $\boldsymbol{\pi}$, uniform $P$). We report the Spearman rank correlation of per-instance $\eta$ with the default ($\rho$), the mean absolute deviation in $\eta$ ($|\Delta\eta|$), and the mean $\eta$ under the variant.

| Variant | $\rho$ | $|\Delta\eta|$ | Mean $\eta$ |
|---|---|---|---|
| Text-proximity $P$ | 0.993 | 0.009 | 0.149 |
| Exp-decay $P$ | 0.968 | 0.018 | 0.145 |
| Inverse-fan-in $P$ | 0.964 | 0.028 | 0.130 |
| Subtree-weighted $P$ | 0.861 | 0.062 | 0.218 |
| Recency $\boldsymbol{\pi}$ | 0.850 | 0.118 | 0.276 |
| Recency $\boldsymbol{\pi}$ + Text-prox. $P$ | 0.847 | 0.109 | 0.267 |
| Degree $\boldsymbol{\pi}$ | 0.778 | 0.074 | 0.228 |

### A.3. Sensitivity to Extraction Granularity

To assess how extraction errors propagate to $\eta$, we apply controlled perturbations to each extracted graph, removing a single node or edge, or adding 1-4 random edges, and report the coefficient of variation of $\eta$ by puzzle size. Sensitivity decreases with graph size; for $6 \times 6$ graphs and larger, single-perturbation CV is below $5\%$.

*Table 6.* **Sensitivity of $\eta$ to graph perturbations.** Coefficient of variation of $\eta$ under each perturbation type, by puzzle size.

| Perturbation | $4 \times 4$ | $5 \times 5$ | $6 \times 6$ | $7 \times 7$ |
|---|---|---|---|---|
| Node removal | 0.130 | 0.079 | 0.051 | 0.037 |
| Edge removal | 0.115 | 0.060 | 0.043 | 0.035 |
| Edge add (1) | 0.165 | 0.074 | 0.051 | 0.038 |
| Edge add (2) | 0.260 | 0.107 | 0.075 | 0.052 |
| Edge add (3) | 0.300 | 0.134 | 0.091 | 0.071 |
| Edge add (4) | 0.340 | 0.167 | 0.118 | 0.081 |

## B. Environment and Claim Verification

For each puzzle family, we implement deterministic verifiers that map an extracted claim to a correctness label (correct / incorrect), using the executable environment. Placement claims are checked against the environment's ground-truth solution, while constraint claims are checked against the given puzzle state (clues). State-dependent (non-static) claims are not verified as tracking partial solutions is not trivial and not robust. If a branch is explicitly presented as a contradiction in the trace, we do not count the triggering claim as incorrect (it is used as part of a refutation). Because this logic relies on faithful extraction of commitments and branch boundaries, we use it conservatively to avoid introducing fragility.

**Edge validation.** Whereas claim nodes are verified deterministically against the environment, edges are attributed by constrained LLM rule application and are not programmatically checked. To quantify edge quality, we manually evaluated 200 randomly sampled rule applications: 151 (75.5%) were fully correct under a strict criterion in which a single missing premise counts as a full error. Most errors were incomplete derivations (a missing premise) rather than spurious edges.

**Unverifiable claims.** State-dependent (non-static) claims reference the current partial board state and are excluded from verification to avoid cascading grading errors. Table 7 reports the fraction of such claims per model and puzzle size; all fractions are below 15%. Because errors in these claims typically propagate to later verifiable claims, their practical effect is a slightly delayed first-error detection and a mild attenuation of the $\eta$-accuracy correlation.

## C. Structural Analysis Details

The structural analysis in Figure 5 and Figure 6 is computed on 85 solved *Tents* instances (3 traces $\times$ 3 instances per size per model; 20/35/30 for DeepSeek/Kimi/Qwen3), balanced across sizes (24/20/21/20 for $4 \times 4$ through $7 \times 7$).

*Table 7.* **Fraction of unverifiable (state-dependent) claims.** Mean ± std by model and puzzle size.

| Model | $4 \times 4$ | $5 \times 5$ | $6 \times 6$ | $7 \times 7$ |
|---|---|---|---|---|
| Qwen3 | $0.00 \pm 0.00$ | $0.13 \pm 0.16$ | $0.11 \pm 0.11$ | $0.07 \pm 0.08$ |
| DeepSeek | $0.03 \pm 0.06$ | $0.06 \pm 0.10$ | $0.07 \pm 0.07$ | $0.14 \pm 0.14$ |
| Kimi | $0.00 \pm 0.00$ | $0.13 \pm 0.19$ | $0.08 \pm 0.09$ | $0.14 \pm 0.17$ |

**Token-quality correlations within model and size.** To rule out that the near-zero token-$\eta$ relationship is an artifact of pooling across difficulty, we stratify by model and by puzzle size (Table 8). Token-$\eta$ correlations are near zero at every level, while token-accuracy correlations are consistently negative and strengthen with size, consistent with token count proxying for difficulty rather than reasoning quality (Muennighoff et al., 2025; Shojaee et al., 2025).

*Table 8.* **Token-quality correlations, stratified.** Pearson correlations of completion tokens with $\eta$ and with accuracy, within each model and within each puzzle size.

| Stratum | Tokens vs. $\eta$ | Tokens vs. Acc. |
|---|---|---|
| DeepSeek | $-0.399$ | $-0.756$ |
| Kimi | $+0.176$ | $-0.388$ |
| Qwen3 | $-0.397$ | $-0.770$ |
| $4 \times 4$ | $+0.020$ | $-0.324$ |
| $5 \times 5$ | $-0.028$ | $-0.604$ |
| $6 \times 6$ | $-0.032$ | $-0.532$ |
| $7 \times 7$ | $-0.040$ | $-0.771$ |

**Per-Model Structural-Metric Correlations** Table 2 reports correlations pooled across models; here we give the per-model breakdown. The directional trends are consistent across all three source models: $|V|$, width, and token count correlate *negatively* with accuracy for every model, while $\eta$ is the only metric that correlates *positively* with accuracy throughout. The associations with $\eta$ (right-hand columns) are weaker and vary in sign across models, but every significant $\eta$ correlation is negative and all sign-flipped (positive) values, notably Kimi's entire column, are non-significant.

*Table 9.* **Per-model structural-metric correlations.** Pearson correlations of each structural metric with claim accuracy and with $\eta$, by source model (cf. the pooled values in Table 2). Significance: $^{*}p < 0.05$, $^{**}p < 0.01$, $^{***}p < 0.001$.

| Metric | DeepSeek V3.2 | | Kimi K2 | | Qwen3 235B | |
|---|---|---|---|---|---|---|
| | Acc. | $\eta$ | Acc. | $\eta$ | Acc. | $\eta$ |
| Depth | $-0.226$ | $-0.166$ | $-0.403^{*}$ | $+0.272$ | $-0.225$ | $+0.034$ |
| Diameter | $-0.342$ | $-0.233$ | $-0.394^{*}$ | $+0.158$ | $-0.335$ | $-0.007$ |
| Avg. path length | $-0.046$ | $-0.165$ | $-0.296$ | $+0.076$ | $-0.192$ | $+0.156$ |
| Width | $-0.903^{***}$ | $-0.472^{*}$ | $-0.483^{**}$ | $+0.088$ | $-0.667^{***}$ | $-0.545^{**}$ |
| $|V|$ | $-0.899^{***}$ | $-0.492^{*}$ | $-0.525^{**}$ | $+0.104$ | $-0.708^{***}$ | $-0.551^{**}$ |
| Tokens | $-0.838^{***}$ | $-0.399$ | $-0.450^{**}$ | $+0.176$ | $-0.770^{***}$ | $-0.397^{*}$ |
| $\eta$ | $+0.440$ | — | $+0.174$ | — | $+0.456^{*}$ | — |

**Efficiency below the solved subset.** We additionally compare structural metrics on *incorrect* Tents traces against correct traces at matched puzzle sizes. We exclude failed *generations* (truncated or aborted runs producing fewer than $\sim 1\text{k}$ completion tokens and no extractable reasoning), which reflect infrastructure failures rather than flawed reasoning. On the remaining failed traces ($n = 10$ incorrect vs. 35 matched correct, sizes $5 \times 5$–$7 \times 7$, three models; Table 10), reasoning-flow efficiency is $57\%$ lower, claim accuracy is $\sim 14\%$ lower, and the first error appears $\sim 51\%$ earlier. Failed traces are also larger ($+52\%$ nodes), wider ($+73\%$), and higher-entropy, and devote a smaller fraction of the graph to the solution, indicating that failing models reason more diffusely from early on rather than making isolated late mistakes. This analysis is limited to Tents; a broader study across puzzle families is left to future work.

**Bootstrap confidence intervals.** For every association involving $\eta$ in Figure 5, we compute bootstrap $95\%$ confidence intervals (5,000 resamples over the 85 instances above). All intervals exclude zero except the one for token count, consistent with $\eta$ being uncorrelated with verbosity but reliably associated with the other quantities: token count ($r = -0.05$,

*Table 10.* **Correct vs. failed traces (Tents).** Pooled means over matched correct and genuinely failed traces; failed *generations* ($< 1$k tokens, no extractable reasoning) are excluded. $\eta$ uses the same (solution-excluded) convention as the main analysis.

| Metric | Correct | Failed | $\Delta$ |
|---|---|---|---|
| Claim accuracy | 0.90 | 0.78 | $-14\%$ |
| First-error depth (norm.) | 0.37 | 0.18 | $-51\%$ |
| Efficiency $\eta$ | 0.127 | 0.055 | $-57\%$ |
| Solution fraction $|V_{\text{sol}}|/|V|$ | 0.43 | 0.33 | $-23\%$ |
| Graph size $|V|$ | 168 | 256 | $+52\%$ |
| Width | 113 | 195 | $+73\%$ |
| Flow entropy $H_{\text{str}}$ | 4.84 | 5.24 | $+8\%$ |

$[-0.23, 0.13]$), solution-supporting fraction ($r = 0.55$, $[0.40, 0.66]$), claim accuracy ($r = 0.33$, $[0.16, 0.49]$), first-error depth ($r = 0.28$, $[0.03, 0.50]$), restatements per claim ($r = 0.27$, $[0.03, 0.50]$), and graph size ($r = -0.33$, $[-0.48, -0.14]$).

# D. Detailed performance

This section reports per-puzzle accuracy (solved / total) and mean completion tokens (in parentheses) for each difficulty level, complementing the aggregate benchmark table in the main text (Table 1). The tables below break results down by model family.

*Table 11.* **Detailed performance (GPT-5).** Per-puzzle results by difficulty on the 21-puzzle suite. Each cell reports # solved / # evaluated instances, with mean completion token count in parentheses, computed over the evaluated instances at that difficulty. All runs use the same solver prompt and decoding settings as in the main benchmark (Section 4.1).

| Puzzle | Trivial | Easy | Normal | Hard |
|---|---|---|---|---|
| Solo | 5/5 (1270.8) | 0/5 (798.2) | 5/5 (7250.4) | 4/5 (17960.2) |
| Tents | 5/5 (3165.2) | 5/5 (7402.8) | 5/5 (13974.8) | 0/5 (22825.3) |
| Filling | 5/5 (3957.6) | 1/5 (24751.2) | 0/5 (24745.7) | 0/5 (16066.4) |
| Lightup | 5/5 (1657.0) | 5/5 (7586.8) | 5/5 (16345.8) | 0/5 (29066.4) |
| Mosaic | 5/5 (1205.6) | 5/5 (3708.0) | 4/5 (21926.0) | 0/5 (8904.4) |
| Keen | 5/5 (2319.2) | 5/5 (5098.0) | 5/5 (13839.6) | 0/5 (40029.6) |
| Range | 5/5 (3075.0) | 5/5 (8830.0) | 4/5 (24656.6) | 0/5 (24439.8) |
| Pattern | 5/5 (632.4) | 5/5 (27481.2) | 0/5 (25613.6) | 0/5 (13722.2) |
| Dominosa | 5/5 (1316.8) | 5/5 (10460.4) | 5/5 (39934.4) | 0/5 (21611.0) |
| Galaxies | 5/5 (8990.8) | 2/5 (21186.0) | 0/5 (17832.8) | 0/5 (5402.2) |
| Magnets | 5/5 (3696.6) | 5/5 (13729.2) | 5/5 (14493.6) | 0/5 (3365.2) |
| Net | 5/5 (2937.0) | 0/5 (9123.2) | 0/5 (13990.4) | 0/5 (9220.2) |
| Pearl | 5/5 (13059.0) | 5/5 (15109.4) | 0/5 (41637.2) | 0/5 (20713.7) |
| Rect | 0/5 (500.4) | 5/5 (6514.0) | 4/5 (14525.6) | 0/5 (25371.6) |
| Signpost | 3/5 (436.4) | 1/5 (3173.8) | 2/5 (10093.4) | 0/5 (29715.8) |
| Singles | 5/5 (1413.2) | 5/5 (5431.8) | 5/5 (9047.0) | 1/5 (17037.2) |
| Towers | 5/5 (1948.0) | 5/5 (7965.2) | 5/5 (13068.0) | 0/5 (27154.0) |
| Tracks | 0/5 (17736.0) | 0/5 (4303.8) | 0/5 (7751.8) | 0/5 (1954.0) |
| Undead | 5/5 (7803.0) | 4/5 (16251.6) | 5/5 (11390.0) | 1/5 (25238.4) |
| Unequal | 5/5 (2856.8) | 5/5 (4854.8) | 3/5 (7384.4) | 0/5 (30060.4) |
| Unruly | 0/5 (7246.8) | 0/5 (10013.2) | 0/5 (13245.4) | 0/5 (27242.4) |

# E. Prompts

This appendix collects the prompts used in our pipeline: the solver prompt used to elicit puzzle solutions, the prompts used for claim extraction and screening, the prompts used for rule extraction, and the puzzle-specific claim and rule schemas.

## E.1. Solver Prompt

Figure 7 displays the prompt used to solve the puzzles.

## E.2. Claim Extraction Prompts

This section presents the prompts used to extract the set of claims found in the model's reasoning trace. The first prompt pair (Figure 8) determines whether the trace assumes 0-based or 1-based indexing, ensuring consistent interpretation of

*Table 12.* **Detailed performance (Qwen3 235B).** Per-puzzle results by difficulty on the 21-puzzle suite. Each cell reports # solved / # evaluated instances, with mean completion token count in parentheses, computed over the evaluated instances at that difficulty. All runs use the same solver prompt and decoding settings as in the main benchmark (Section 4.1).

| Puzzle | Trivial | Easy | Normal | Hard |
|---|---|---|---|---|
| Solo | 5/5 (5220.4) | 0/5 (19878.2) | 0/5 (29829.8) | 0/5 (29961.8) |
| Tents | 5/5 (6486.0) | 5/5 (11591.6) | 4/5 (21452.4) | 0/5 (19693.3) |
| Filling | 5/5 (8260.6) | 0/5 (24198.3) | 0/5 (24059.6) | 0/5 (22894.2) |
| Lightup | 5/5 (6770.0) | 5/5 (13369.6) | 1/5 (18392.4) | 0/5 (26117.4) |
| Mosaic | 5/5 (6223.8) | 5/5 (14351.0) | 0/5 (24837.8) | 0/5 (17945.6) |
| Keen | 5/5 (4829.0) | 5/5 (12328.0) | 2/5 (24928.2) | 0/5 (25493.4) |
| Range | 5/5 (10886.8) | 4/5 (20025.2) | 0/5 (21463.8) | 0/5 (22451.6) |
| Pattern | 5/5 (3589.8) | 0/ 5 (27557.2) | 0/5 (25310.2) | 0/5 (21853.8) |
| Dominosa | 5/5 (4632.8) | 3/5 (18389.8) | 0/5 (21440.4) | 0/5 (20329.6) |
| Galaxies | 2/5 (17782.4) | 0/5 (22570.0) | 0/5 (23114.8) | 0/5 (25084.6) |
| Magnets | 3/5 (15607.8) | 3/5 (28395.8) | 2/5 (26639.0) | 0/5 (21861.6) |
| Net | 5/5 (10330.4) | 0/5 (27653.0) | 0/5 (31310.0) | 0/5 (23810.4) |
| Pearl | 1/5 (19109.4) | 0/5 (20671.2) | 0/5 (19713.0) | 0/5 (18234.4) |
| Rect | 0/5 (1787.8) | 0/5 (21397.6) | 0/5 (25064.4) | 0/5 (22270.2) |
| Signpost | 3/5 (2826.2) | 1/5 (7387.6) | 2/5 (20081.6) | 0/5 (23929.8) |
| Singles | 5/5 (4112.6) | 5/5 (11495.8) | 5/5 (13541.8) | 0/5 (24158.975) |
| Towers | 5/5 (9815.6) | 5/5 (18604.0) | 4/5 (23485.64) | 0/5 (31988.6) |
| Tracks | 0/5 (25576.2) | 0/5 (20408.0) | 0/5 (20679.6) | 0/5 (23823.4) |
| Undead | 1/5 (26931.2) | 1/5 (25282.0) | 0/5 (30382.4) | 0/5 (24819.8) |
| Unequal | 3/5 (6154.0) | 5/5 (10992.4) | 2/5 (15559.4) | 0/5 (27065.8) |
| Unruly | 0/5 (18470.6) | 0/5 (23148.6) | 0/4 (23897.6) | 0/5 (21991.36) |

```
You are a logic puzzle expert. You will be given a logic puzzle to solve. Here is a description of the puzzle:
{PUZZLE DESCRIPTION}
You receive an array representation of the puzzle state as a grid. Your task is to solve the puzzle by filling out the grid with
the correct values. You need to solve the puzzle on your own, you cannot use any external resources or run any code. Once you
have solved the puzzle, tell me the final answer without explanation. Return the final answer as a JSON array of arrays.
Here is the current state of the puzzle as a string of the internal state representation:
{PUZZLE STATE}
```

*Figure 7.* Prompt for solving the instance described in the puzzle state. Puzzle descriptions are detailed in Section E.5.

references. The second prompt pair (Figure 9) extracts atomic claims from the raw reasoning trace using a pure LLM-based approach. The third prompt pair (Figure 10) augments the claims that are extracted by the regex by recovering missing or implicit statements that are difficult to capture with rules alone. The fourth prompt pair (Figure 11) filters, normalizes, and removes redundant or ill-formed claims to produce a clean claim set.

```
You are a grid specialist. Your task is to check if the indexing used in the text you get is 0-based or 1-based. All you return
is either the word "0-based" or "1-based", nothing else.
```

```
Here is the text: {TRACE}
```

*Figure 8.* System (top) and user (bottom) prompts for determining whether the reasoning trace relies on 0-based or 1-based indexing.

### E.3. Rule Extraction Prompts

The rule extraction prompt (Figure 12) identifies deduction rules between claims, which define the directed edges of the reasoning graph (between premises and conclusions).

### E.4. Puzzle-specific Claims and Rules

These prompts are puzzle-specific and define the type of rules and claims that the model should extract for a given puzzle. The claim and rule types for *Tents* are shown in Figure 13 and Figure 14, and for Sudoku in Figure 15 and Figure 14, respectively.

*Table 13.* **Detailed performance (Kimi K2 Thinking).** Per-puzzle results by difficulty on the 21-puzzle suite. Each cell reports # solved / # evaluated instances, with mean completion token count in parentheses, computed over the evaluated instances at that difficulty. All runs use the same solver prompt and decoding settings as in the main benchmark (Section 4.1).

| Puzzle | Trivial | Easy | Normal | Hard |
|---|---|---|---|---|
| Solo | 5/5 (2302.4) | 0/5 (45200) | 4/5 (30893.4) | 1/5 (56261.4) |
| Tents | 5/5 (5734.4) | 5/5 (16160.8) | 5/5 (28705.2) | 0/5 (64210.8) |
| Filling | 5/5 (8910.4) | 0/5 (46002.2) | 0/5 (49713.7) | 0/5 (60694.4) |
| Lightup | 5/5 (2336.6) | 5/5 (12332.8) | 5/5 (31801.6) | 0/5 (65536.0) |
| Mosaic | 5/5 (7021.2) | 5/5 (8693.6) | 2/5 (54524.0) | 0/5 (59351.2) |
| Keen | 5/5 (2894.2) | 5/5 (13424.8) | 4/5 (33754.4) | 0/4 (62941.0) |
| Range | 5/5 (5850.2) | 4/5 (20867.0) | 0/5 (57348.8) | 0/5 (58553.2) |
| Pattern | 5/5 (1438.6) | 2/5 (55358.6) | 0/5 (65138.0) | 0/5 (59118.2) |
| Dominosa | 5/5 (2134.0) | 5/5 (30288.2) | 0/5 (65536.0) | 0/5 (64695.4) |
| Galaxies | 4/5 (28160.8) | 2/5 (56873.0) | 0/5 (55219.6) | 0/3 (51139.3) |
| Magnets | 5/5 (7288.2) | 4/5 (33953.6) | 5/5 (34259.2) | 0/5 (65536.0) |
| Net | 5/5 (9660.4) | 0/5 (25680.8) | 0/5 (44706.2) | 0/5 (59358.8) |
| Pearl | 4/5 (24180.4) | 1/5 (53557.8) | 0/5 (62605.2) | 0/5 (64270.6) |
| Rect | 0/5 (1802.6) | 2/5 (16901.0) | 1/5 (27804.4) | 0/5 (65536.0) |
| Signpost | 3/5 (1662.0) | 1/5 (8682.0) | 1/5 (32537.4) | 0/5 (64912.2) |
| Singles | 5/5 (2120.4) | 5/5 (9441.0) | 5/5 (17775.8) | 0/5 (65536.0) |
| Towers | 5/5 (4798.4) | 5/5 (17015.6) | 5/5 (43610.4) | 0/5 (51806.02) |
| Tracks | 1/5 (45695.2) | 0/5 (64871.4) | 0/5 (62114.0) | 0/5 (56650.4) |
| Undead | 2/5 (35891.4) | 4/5 (34211.8) | 2/5 (48413.0) | 0/5 (61080.2) |
| Unequal | 2/5 (3252.6) | 4/5 (22244.4) | 4/5 (16272.8) | 0/5 (64726.2) |
| Unruly | 0/5 (19496.6) | 0/5 (32227.6) | 0/5 (56042.2) | 0/5 (65536.0) |

## E.5. Puzzle Descriptions

Information and details specific to each puzzle are presented in Table 15. Below, we provide the full textual descriptions for each puzzle used in our suite, which define the rules and expected solution format.

*Table 14.* **Detailed performance (DeepSeek V3.2).** Per-puzzle results by difficulty on the 21-puzzle suite. Each cell reports # solved / # evaluated instances, with mean completion token count in parentheses, computed over the evaluated instances at that difficulty. All runs use the same solver prompt and decoding settings as in the main benchmark (Section 4.1).

| Puzzle | Trivial | Easy | Normal | Hard |
|---|---|---|---|---|
| Solo | 5/5 (4261.0) | 0/5 (19392.0) | 5/5 (39878.0) | 0/5 (65536.0) |
| Tents | 4/5 (3890.0) | 5/5 (9681.6) | 5/5 (19950.6) | 0/5 (28922.4) |
| Filling | 5/5 (7543.4) | 0/5 (26743.1) | 0/5 (25723.0) | 0/5 (31961.8) |
| Lightup | 5/5 (2930.4) | 5/5 (11211.2) | 5/5 (21151.2) | 0/5 (29290.4) |
| Mosaic | 3/5 (1601.2) | 4/5 (5567.6) | 5/5 (21151.2) | 0/5 (24473.2) |
| Keen | 5/5 (3156.4) | 5/5 (8420.2) | 5/5 (18802.4) | 0/5 (46462.0) |
| Range | 5/5 (7486.8) | 4/5 (19574.2) | 1/5 (25182.7) | 0/5 (33962.6) |
| Pattern | 5/5 (2696.8) | 2/5 (48120.0) | 0/5 (33097.8) | 0/5 (33757.2) |
| Dominosa | 5/5 (3100.0) | 4/5 (23104.8) | 0/5 (34934.6) | 0/5 (35696.4) |
| Galaxies | 3/5 (18515.6) | 3/5 (29391.4) | 0/5 (27128.0) | 0/5 (25761.8) |
| Magnets | 5/5 (8124.2) | 5/5 (25038.8) | 4/5 (31377.8) | 0/5 (32475.2) |
| Net | 3/5 (5812.2) | 0/5 (29112.6) | 0/5 (21182.0) | 0/5 (19381.8) |
| Pearl | 3/5 (20979.2) | 0/5 (35024.6) | 0/5 (36094.2) | 0/5 (32317.2) |
| Rect | 0/5 (1965.8) | 1/5 (10508.2) | 1/5 (28211.2) | 0/5 (36560.4) |
| Signpost | 3/5 (1489.6) | 0/5 (8036.6) | 2/5 (27705.2) | 0/5 (39522.6) |
| Singles | 5/5 (2930.0) | 5/5 (10367.2) | 5/5 (14680.4) | 0/5 (36012.4) |
| Towers | 5/5 (7000.6) | 5/5 (16444.8) | 5/5 (36753.5) | 0/5 (64782.1) |
| Tracks | 2/5 (21102.2) | 0/5 (31926.6) | 0/5 (29741.8) | 0/5 (17576.6) |
| Undead | 5/5 (15865.4) | 5/5 (24152.0) | 2/5 (27259.0) | 0/5 (36559.2) |
| Unequal | 5/5 (3756.2) | 3/5 (11294.8) | 2/5 (13527.8) | 0/5 (62481.5) |
| Unruly | 0/5 (17382.0) | 0/5 (30172.4) | 0/5 (34259.1) | 0/5 (39043.4) |

```
You are a deterministic information extraction engine for reasoning traces solving a {PUZZLE} instance.
TASK
Extract ALL claims from scratch that are directly supported by the reasoning trace segment.
RULES
- Use only what is stated or explicitly concluded in the reasoning trace segment.
- Track branches: each new hypothetical assumption (\assume/suppose/if...") starts the next BRANCH#<int> in order of appearance.
Make sure to use the tentative status and branch scope appropriately.
- Claims inside a branch must use scope="BRANCH#k".
- When the reasoning returns to mainline after a tentative claim and states a conclusion, that claim must be in scope="MAIN".
- Do extract claims that are mentioned as premises (\since/given/using ..."); such premise mentions are valid claim occurrences
and should be extracted as separate claims.
- Do NOT compute implied consequences (e.g., {IMPLIED_CONSEQUENCE_EXAMPLE}) unless the reasoning trace explicitly states them.
- If the same claim content is asserted multiple times in different sentences, output multiple claim objects (one per assertion
event). Do NOT deduplicate.
{CLAIM_DESCRIPTION}
PREMISE-USE EXTRACTION (MANDATORY)
- If a sentence explicitly *uses* a fact as a premise (markers: "since", "given", "using", "because", "therefore", "thus",
"hence", "so"), then extract each referenced claim occurrence that is explicitly mentioned in that sentence.
- Do NOT treat generic words like "as" or "from" as premise markers.
- A single sentence may contain multiple claim occurrences; output one claim object per occurrence, each with source_sents=["S#",
...].
Ignore any coordinates appearing in:
- examples,
- rule descriptions,
- schema text,
- meta-instructions.
A raw grid or JSON array by itself is NOT an assertion event unless the reasoning trace explicitly states that it is the final
solution and explains the symbol-to-meaning mapping.
Only extract claims from the reasoning trace segment sentences (S#).
{PUZZLE_SPECIFIC_NOTES}
If no source_sents sentence contains a valid claim occurrence marker (assertion OR premise use), REMOVE the claim.
FINAL OUTPUT FORMAT
[
    {{"kind": string, "payload": dict, "status": string, "scope": string, "source_sents": [string, ...]}},
]
where kind is the claim kind, payload is the claim data as per above, status/scope/source_sents are the metadata fields as per
above.
```

```
Now, extract EVERY claim made in the reasoning trace given further below. Make sure to follow the previously given rules \&
specifications.
The coordinates in the reasoning trace are {INDEXING}. Note that the output should always use 0-based coordinates.
REASONING TRACE:
{TRACE}
Extract claims as a JSON list only.
```

*Figure 9.* System (top) and user (bottom) prompts for extracting the claims from the reasoning trace.

```
You are a deterministic information extraction engine for reasoning traces solving a {PUZZLE} instance.
TASK
You will be given:
- a reasoning trace segment (sentences labeled S1, S2, ...),
- a draft list of extracted claims (JSON)
Your job: COMPLEMENT the draft claims.
DO
- Fix formatting/schema issues so every claim matches the allowed kinds and payload schemas.
- Ensure every claim has required metadata: kind, payload, status, scope, source_sents (NON-EMPTY).
- Add any clearly missing claims that are directly supported by the reasoning trace segment.
- Preserve assertion events: if the same claim content appears as multiple asserted events in different sentences, keep them as
separate claim objects (do not merge).
DO_NOT
- Do not invent claims not supported by the reasoning trace.
- Do not \compute" new claims unless the reasoning trace explicitly states the conclusion (e.g., {COMPUTED_CLAIM_EXAMPLE}).
- Do not add claims that are only logical consequences (e.g., {IMPLIED_CONSEQUENCE_EXAMPLE}) unless the reasoning trace
explicitly asserts them.
- Do NOT repair a claim by guessing missing coordinates or payload values. If repairing would require guessing, drop the claim
(or mark tentative per above, but only if the claim is truely tentative).
{COMMON_ERRORS}
Never delete any draft claim UNLESS:
- it is schema-invalid and cannot be repaired without guessing payload fields, OR
- it has no possible asserting sentence in the provided TRACE.
If a draft claim has no valid claim-occurrence marker in its listed source_sents, DELETE it.
Do NOT "rescue" unsupported claims by marking tentative unless the sentence explicitly contains hypothetical language
("assume/suppose/if").
{CLAIM_DESCRIPTION}
PREMISE-USE EXTRACTION (MANDATORY)
- If a sentence explicitly *uses* a fact as a premise (markers: "since", "given", "using", "because", "therefore", "thus",
"hence", "so"), then extract each referenced claim occurrence that is explicitly mentioned in that sentence.
- Do NOT treat generic words like "as" or "from" as premise markers.
- A single sentence may contain multiple claim occurrences; output one claim object per occurrence, each with source_sents=["S#",
...].
Ignore any coordinates appearing in:
- examples,
- rule descriptions,
- schema text,
- meta-instructions.
A raw grid or JSON array by itself is NOT an assertion event unless the reasoning trace explicitly states that it is the final
solution and explains the symbol-to-meaning mapping.
Only extract claims from the TRACE segment sentences (S#).
FINAL OUTPUT FORMAT
[
    {{"kind": string, "payload": dict, "status": string, "scope": string, "source_sents": [string, ...]}},
]
where kind is the claim kind, payload is the claim data as per above, status/scope/source_sents are the metadata fields as per
above.
```

```
Now, complement the draft claims and make sure to fix the metadata. Further add new claims that are missing, such that EVERY
claim is extracted from the reasoning trace.
The coordinates in the trace are {INDEXING}. Note that the output should always use 0-based coordinates.
REASONING TRACE:
{TRACE}
DRAFT CLAIMS (JSON):
{CLAIMS}
Return the complemented claims as a JSON list only.
```

*Figure 10.* System (top) and user (bottom) prompts for complementing claims extracted with a rule-based method.

```
You are a strict validator for extracted claims from a reasoning trace solving a {PUZZLE} instance.
TASK
You will be given:
- the reasoning trace (sentences labeled S1, S2, ...),
- a combined list of candidate claims (JSON)
Your job: SCREEN AND FILTER.
SEMANTICS (CRITICAL)
- A claim is VALID if at least one sentence in the reasoning trace CONTAINS A CLAIM OCCURRENCE.
- A claim occurrence may be:
    (a) an explicit assertion or conclusion, OR
    (b) a restatement or summary, OR
    (c) an assumption or hypothetical statement (should have the tentative status if this is the case), OR
    (d) a premise use / reminder (e.g., \since / given / using ...").
- Claims must NOT be inferred beyond what is explicitly written.
- `source_sents` must contain ONLY sentences that actually contain the claim occurrence (assertion, restatement, or premise use).
DO
- Remove any claim for which NO sentence in the reasoning trace contains the claim occurrence.
- Enforce schemas and metadata requirements exactly.
- Fix minor normalization issues when unambiguous.
- Keep multiple identical claims if they correspond to DISTINCT sentence occurrences.
- Do NOT suppress claims because they are redundant, obvious, unused later, or logically implied by other claims.
- If a claim has multiple source_sents, REMOVE any sentence that does not itself contain the claim occurrence.
- Make sure that tentative claims and claims based on a tentative assumption are marked status="tentative" and scope="BRANCH#k"
appropriately and then merged into main properly.
DO_NOT
- Do not add new claims (this stage is not for recall).
- Do not compute implied consequences.
- Do not \upgrade" or \downgrade" status unless the sentence explicitly indicates hypothetical vs non-hypothetical language.
DEDUPLICATION (STRICT AND MECHANICAL)
- Deduplicate claims ONLY if they are EXACTLY identical in: (kind, payload, status, scope, source_sents).
- In that case, keep one copy and discard the rest.
- Do NOT merge or union source_sents across different sentences.
- Do NOT deduplicate claims that originate from different sentences, even if kind/payload are identical.
- Do NOT deduplicate across scopes (e.g., MAIN vs BRANCH#k).
PREMISE-USE EXTRACTION (MANDATORY)
If a sentence contains any of these: "since", "given", "using", "because", "as", "from", then you MUST keep every correctly
referenced claim of any kind mentioned in that sentence, even if it was stated earlier.
{CLAIM_DESCRIPTION}
OUTPUT (STRICT)
- Output ONLY a JSON list.
- No markdown, no commentary, no extra keys.
```

```
Now, validate  the extracted claims. Make sure to not filter out repeated claims if they are made in different sentences, and
make sure to keep all of the valid claims as in the previously given rules.
The coordinates in the reasoning trace are {INDEXING}. Note that the output should always use 0-based coordinates.
REASONING TRACE:
{TRACE}
CANDIDATE CLAIMS (JSON):
{CLAIMS}
Return the screened + unified claims as a JSON list only.
```

*Figure 11.* System (top) and user (bottom) prompts for filtering and cleaning claims.

```
You are labeling reasoning steps in reasoning trace solving a {PUZZLE} instance.
TASK
You will be given:
- ONE claim to explain (the target claim),
- part of the reasoning trace just before the claim to justify (target claim),
- a list of already-extracted claims available up to that prefix,
Your job: Output AT MOST ONE rule application that explains how the reasoning trace derives that target claim. Output none if the
target claim is not derived.
DO
- You MUST follow the steps below exactly.
- You MUST NOT create new claims.
- You MUST use only the provided claim_ids in inputs/output (except "C_MISSING" allowed only in inputs).
- You MUST output either:
   (a) [] (if no rule clearly applies or if the target claim truely is stated (see note below)), OR
   (b) a JSON with exactly one rule app dict.
RULE APP SCHEMA
Every rule app MUST have exactly these fields:
- rule: one of the allowed rule ids
- inputs: [claim_id, ...]           (The claims used by THIS rule to produce THIS output; use "C_MISSING" for any clearly needed
premise that is not in the provided CLAIM LIST)
- output: claim_id                  (EXACTLY ONE; must be the target claim id)
- support_sents: ["S#", ...]        (NON-EMPTY; sentences that directly justify that THIS rule produced THIS output; this should
be the sentence IDs/Numbers from the reasoning trace or claim support sentences, so S1, S2, ....; do not include the sentence
texts; the input and the support sentences have whole sentences + id, but here only include the ids/numbers)
claim_id format:
- Claim ids are strings like "C1", "C2", ...
- Special id: "C_MISSING" may appear ONLY in inputs to represent a missing premise the reasoning trace clearly relies on.
- "C_MISSING" MUST NEVER be used as output.
{RULE_DESCRIPTION}
GLOBAL CONSTRAINTS
- Wiring-only (NO NEW CLAIMS)
   (a) The output must be an existing claim_id from the provided CLAIM LIST.
   (b) Every non-missing input must be an existing claim_id from the provided CLAIM LIST.
- One-claim-at-a-time: Each app explains exactly ONE output claim: the target claim.
- Minimality
   (a) The app must match EXACTLY what the reasoning trace explicitly concludes for the target claim in support_sents.
   (b) No closure, no implicit consequences, no \and therefore also ...".
- Scope
   (a) For all rules EXCEPT R_TENTATIVE_DISCHARGE: all inputs and the output must have the SAME scope.
   (b) For R_TENTATIVE_DISCHARGE: inputs are from BRANCH#k and output is in MAIN.
- Tentative containment:
   (a) tentative claims MUST have scope BRANCH#k (never MAIN).
   (b) No tentative claim may influence MAIN except via R_TENTATIVE_DISCHARGE.
- Rule choice must be unambiguous
   (a) Choose the SINGLE best matching rule per the rule meanings below.
   (b) If none match, output [] (no app for this target).
- Use all inputs
   (a) Make sure to all inputs listed are actually used by the rule application.
   (b) Make sure to not omit any input that the reasoning trace clearly relies on to reach the target claim.
When you output a rule application, include all premises that the reasoning trace relies on to reach the target claim under that
rule. If the reasoning trace clearly relies on a premise but it is not present in the provided claim list, include "C_MISSING" in
the inputs to represent that missing premise.
DEFINITION: STATED CLAIM
A claim is stated iff the reasoning trace presents it as a primitive puzzle fact or initial condition, without relying on or
referring to any other claims or reasoning steps. Stated claims are not the result of applying any rule in this schema. If a
claim can be explained by any rule (possibly using C_MISSING), it must be treated as derived, regardless of its metadata status.
Returning [] explicitly asserts that the target claim is treated as STATED (i.e., not derivable by any rule).
IMPORTANT - STATED vs DERIVED (Operational Definition)
A claim must be treated as DERIVED if there exists ANY rule in this schema that can reasonably explain it from the reasoning
trace context, even if:
- the claim is pretagged as "stated",
- the reasoning trace restates it as a fact,
- or some required premises are implicit or missing (use C_MISSING).
A claim is treated as STATED if and only if NO rule application in this schema can plausibly explain it, even after considering
the surrounding reasoning trace context and allowing C_MISSING. Using C_MISSING to represent missing premises is encouraged to
find derivations.
Stated claims typically correspond to primitive puzzle facts, but appearance or metadata alone MUST NOT be used to decide this.
Output MUST be valid JSON only. No prose. No markdown.
FINAL OUTPUT FORMAT
Either:
- []
or
- {{"rule": string, "inputs": [string, ...], "output": string, "support_sents": [string, ...]}}
Exactly 0 or 1 app per target claim.
```

```
Now, extract at most one rule application that explains how the reasoning trace derives the target claim, if the claim is
derived, otherwise return []. Make sure to follow the previously given rules \& specifications.
TARGET CLAIM TO EXPLAIN (must be the output of the rule if you produce a rule application):
{CLAIM_TO_EXPLAIN}
REASONING TRACE BEFORE TARGET CLAIM:
{TRACE}
CLAIMS AVAILABLE (claims before target claim, each includes claim_id):
{TRACE_CLAIMS}
Return JSON only:
- [] OR
- {{"rule":..., "inputs":[...], "output":"<target claim_id>", "support_sents":[...]}}
```

*Figure 12.* System (top) and user (bottom) prompts for extracting deduction rules (i.e., edges).

```
CLAIM DESCRIPTION
The core principle of the claims extraction from the reasoning trace is to get a as faithful as possible representation of the
reasoning trace in terms of discrete claims, without adding or removing information beyond what is explicitly written or used in
the trace.
Thus every claim, which corresponds to a claims kind below, must be extracted and preserved as-is in the reasoning trace. Keep
this philosophy in mind when extracting claims. So don't ignore claims, even if they are used at all later on in the reasoning
trace.
We want to capture the reasoning trace as accurately as possible in terms of claims, so we need all claims, even those that seem
irrelevant or unused later on.
And importantly we need to extract all claims, even if they are remembered or restated, they should be a seperate new claim.
Below are the possible kinds of claims that can be made.
ALLOWED CLAIM KINDS + PAYLOAD SCHEMAS
- Tree:           {"r":int,"c":int}
- Quota:          {"axis":"ROW"|"COL","index":int,"tents":int}
- Tent:           {"r":int,"c":int}
- NotTent:        {"r":int,"c":int}
- Remaining:      {"axis":"ROW"|"COL","index":int,"remaining":int}
- TreeCandidate:  {"tree_r":int,"tree_c":int,"cell_r":list[int],"cell_c":list[int]}
- LineCandidates: {"axis":"ROW"|"COL","index":int,"cell_r":list[int],"cell_c":list[int]}
- Contradiction:  {"branch":int,"note":string}
Note the following:
Tree claims are simply assertions that there is a tree at the given cell.
Quota claims assert the total number of tents in a given row or column.
Tent claims assert OR assume that there is a tent at the given cell.
NotTent claims assert that there is NOT a tent at the given cell.
Remaining claims assert how many tents are still to be placed in a given row or column at the current state when it was writen
(i.e. taking into account the claims made so far). Note that maximum tents per row/column is given by the Quota claims. And that
remaining is no inequality, it is an exact number. Further remaining is based on the current state of the trace, so it can go
down as tents are placed, but it can also go up if a tent placement is retracted in a branch.
TreeCandidate claims assert that for the tree at (tree_r, tree_c), the only possible tent placements so far are the listed cells
(cell_r[i], cell_c[i]). And (cell_r[i], cell_c[i]) are ZIPPED coordinate pairs.
LineCandidates claims assert that for the given row or column (axis and index), the only possible tent placements so far are the
listed cells (cell_r[i], cell_c[i]). And (cell_r[i], cell_c[i]) are ZIPPED coordinate pairs.
Contradiction claims assert that there is a contradiction in the trace at the given branch, with an optional note explaining the
contradiction claimed in the reasoning trace.
All coordinates are 0-based indices (i.e., the top-left cell is (0,0)). And axis is either "ROW" or "COL".
Distinction between TreeCandidate and LineCandidates:
- TreeCandidate claims are specific to a particular tree and its possible tent placements.
- LineCandidates claims are specific to a particular row or column and its possible tent placements.
- Do NOT extract TreeCandidate from generic phrases like \only possible", \only option(s)", \only candidate"
  unless the TREE anchor is present in the same sentence or an immediately linked sentence.
- So LineCandidates are typically found together with Remaining or Quota claims (since they are Row/Column oriented), while
TreeCandidate are found together with Tree claims.
- These can also be found together, e.g., first a LineCandidates claim is made for a row, and then a TreeCandidate claim is made
for a specific tree in that row, which narrows down the possible tent placements for that tree.
- Booth are local claims about possible tent placements, but from different perspectives (tree-centric vs line-centric). And they
can inform each other. And further they only use information available at that point in the trace.
- TreeCandidate/LineCandidates represent the set of candidates explicitly asserted (or explicitly implied as 'only') by the trace
at that point, not the objectively correct or complete set. Do not add candidates not mentioned; do not delete candidates not
eliminated in the trace.
- LineCandidates may be asserted even if Remaining is not explicitly stated, as long as the trace explicitly lists the candidate
cells.
Ensure faithfulness to the trace when extracting these claims.
- If the trace lists candidates, record exactly those candidates, even if they appear wrong under the puzzle rules.
- Do not fix adjacency, do not add missing candidates, do not remove candidates unless the trace itself removes them.
Every claim also has some extra metadata fields that you must always include:
- status: "stated"|"derived"|"tentative"
- scope: "MAIN" | "BRANCH#<int>"
- source_sents: ["S#", ...] (NON-EMPTY; sentences that DIRECTLY support the claim, but not necessarily all sentences that MENTION
it or logically imply it. Just the ones that were used as direct evidence for extracting this claim.)
STATUS SEMANTICS
- stated: parameter/fact explicitly given (trees, quotas)
- derived: any non-hypothetical claim occurrence that is not a given parameter, including conclusions AND premise reminders
(\since/given/using ...").
- tentative: hypothetical/assumed (\suppose/assume ...), note that tentative is for the initial assumption, not for conclusions
drawn from it (those are derived)
- Phrases like \we place", \we put", \we choose", \we set" assert derived facts unless explicitly framed as hypothetical.
- If something is assumed for the sake of argument, it is tentative and first if later on confiremed or negated as a new derived
claim in the mainline.
BRANCH SEMANTICS
- MAIN: mainline reasoning
- BRANCH#<int>: numbered branches off the mainline, starting at 1
- Inside branches the status semantics are the same, but the scope is different.
New branches are created by hypothetical reasoning (\suppose", \if", \what if", etc.).
These branches are numbered in order of appearance in the trace and should be tracked consistently for all claims and rules
extracted from that branch.
They end when the reasoning returns to the mainline, often with a conclusion like \therefore", \thus", \hence", etc., or go on to
create sub-branches.
They are most commonly used in proof by contradiction, where a branch is created by assuming something contrary to the mainline,
and then deriving a contradiction within that branch.
So they often end with contradiction claims.
CLAIM OCCURRENCE RULE (CRITICAL)
- Extract a claim whenever the trace explicitly STATES OR RE-STATES OR USES a fact matching a claim kind.
  This includes:
    (a) conclusions/placements (\therefore", \must", \so we place..."),
    (b) restatements/summaries (\so the final grid has ..."),
    (c) premise uses / reminders (\since/given/using ... the tent/tree/quota at ...").
- Each sentence that contains such a claim occurrence produces its own claim object.
  Do NOT deduplicate across different sentences, even if kind/payload are identical.
- Do NOT compute implied consequences that are not explicitly written in the trace.
  Example: Quota(ROW,3)=0 does NOT imply NotTent for every cell in that row unless the trace explicitly says so.
- source_sents include the sentence(s) that contain the occurrence.
  Prefer a SINGLE sentence per claim occurrence whenever possible (one occurrence → one claim object).
If a sentence asserts \tree at (r,c)", it MUST NOT produce Tent or NotTent claims
unless the same sentence explicitly mentions tent placement or exclusion.
Never infer Tent or NotTent from TreeCandidate or LineCandidates claims.
Listing \possible positions", \adjacent cells", or \remaining options" only produces TreeCandidate or LineCandidates, unless the
trace explicitly concludes a placement or exclusion.
Unless it clearly states that there is only one option and that option is chosen/excluded, do NOT extract Tent or NotTent claims
from candidate listings, extract only TreeCandidate or LineCandidates claims.
Also extract tentative tent placements whenever they appear in the reasoning trace, even if they are later retracted or not part
of the final solution, so make sure to extract them AND mark them as tentative.
Such tentative placements are part of the logical process used to conclude whether a tent does or does not exist at a given cell.
In these cases, mark the claim with status="tentative" and assign it to the appropriate branch scope.
```

*Figure 13.* Possible claims for *Tents*.

```
RULE DESCRIPTION
Below are the possible kinds of rule applications that can be made and a description of when and how to use each. Importantly,
use the one rule that matches the reasoning used in the provided reasoning trace to derive the target claim.
Rule applications do NOT create claims; they only label how ONE already-extracted claim (the target claim) was derived in the
reasoning trace, if there was a derivation.
ALLOWED RULE IDS
- R_COUNT_REMAINING
- R_LINE_BLOCK
- R_NO_TOUCH
- R_ONLY_CAND
- R_LINE_CAND
- R_CELL_NEEDS_TREE
- R_CONTRADICTION_CAUSE
- R_CONTRADICTION_DISCHARGE
RULE PARTITIONING (TO AVOID OVERLAP)
Each rule is only allowed to produce certain kinds of outputs:
- R_COUNT_REMAINING:        Remaining only
- R_LINE_BLOCK:             NotTent only
- R_NO_TOUCH:               NotTent only
- R_ONLY_CAND:              TreeCandidate or Tent only
- R_LINE_CAND:              LineCandidates or Tent only
- R_CELL_NEEDS_TREE:        NotTent only
- R_CONTRADICTION_CAUSE:    Contradiction only
- R_TENTATIVE_DISCHARGE:    Tent or NotTent only (in MAIN; negation/confirmation of a tentative assumption)
Tie-breakers for NotTent outputs:
- If NotTent is concluded because a line is full / remaining=0 / quota=0 -> R_LINE_BLOCK
- Else if NotTent is concluded because it would touch a tent -> R_NO_TOUCH
- Else if NotTent is concluded because it has no adjacent tree -> R_CELL_NEEDS_TREE
- Else if it is the MAIN negation/confirmation of a BRANCH tentative assumption -> R_TENTATIVE_DISCHARGE
- Else -> []
RULE MEANINGS (ESSENCE + MINIMAL INPUTS)
1) R_COUNT_REMAINING  (output: Remaining)
Essence:
- The reasoning trace explicitly computes an EXACT remaining count for a row/col by counting placed tents and subtracting from
quota.
Use only if:
- \remaining/left/still to place" language + explicit counting/arithmetic.
Minimal inputs:
- the relevant Quota(...)
- the Tent claims explicitly counted, if they exist; otherwise allow C_MISSING.
Never use for:
- \at most/at least" bounds or contradiction-only arguments.
2) R_LINE_BLOCK  (output: NotTent)
Essence:
- The reasoning trace concludes this cell is NotTent BECAUSE the line has no capacity left (full / 0 remaining / quota 0).
Minimal inputs:
- Remaining(...,0) or Quota(...,0) when available; otherwise C_MISSING if fullness is explicitly asserted.
Never use for:
- no-touch or needs-tree reasoning.
3) R_NO_TOUCH  (output: NotTent)
Essence:
- The reasoning trace concludes this cell is NotTent because it would touch a specific tent (\tents can't touch").
Minimal inputs:
- the causing Tent(...) if available; otherwise C_MISSING if the reasoning trace relies on it.
Never use for:
- needs-tree or fullness reasoning.
4) R_ONLY_CAND  (output: TreeCandidate OR Tent)
Essence:
- Tree-anchored candidate restriction (\for the tree at ... only possible cells are ...") and optionally tree-forced placement.
Use only if:
- reasoning is explicitly anchored to a specific tree.
Minimal inputs:
- Tree(...) for that tree.
- include elimination reasons ONLY if explicitly cited; otherwise omit; use C_MISSING if clearly relied upon but missing.
Tent output allowed only if:
- reasoning trace explicitly concludes/places (\must be", \place/put/set", \so it is").
5) R_LINE_CAND  (output: LineCandidates OR Tent)
Essence:
- Line-anchored candidate restriction (\in row/col ... only possible cells are ...") and optionally placement from line
uniqueness/intersection.
Use only if:
- reasoning is explicitly anchored to a row/col (or intersection of row/col candidates).
Minimal inputs:
- For LineCandidates output: Quota(...) or Remaining(...) for that line when available; otherwise C_MISSING if explicitly relied
upon.
- For Tent output: at least one LineCandidates(...) used for the placement if it exists; otherwise C_MISSING if the reasoning
trace relies on \only possible cell" facts but the needed candidate claim is missing.
Tent output allowed only if:
- reasoning trace explicitly concludes/places (\must be", \place/put/set", \so it is").
6) R_CELL_NEEDS_TREE  (output: NotTent)
Essence:
- The reasoning trace concludes this cell is NotTent because a tent must be adjacent to a tree, and this cell has no adjacent
tree (as argued).
Minimal inputs:
- referenced Tree(...) claims if cited; otherwise omit; use C_MISSING if adjacency-to-tree reasoning is cited but missing in
claims.
Never use for:
- no-touch or fullness.
7) R_CONTRADICTION_CAUSE  (output: Contradiction)
Essence:
- The reasoning trace explicitly declares a contradiction in a BRANCH and cites (at least implicitly) the statements that cannot
all hold.
Use only if:
- the reasoning trace says \contradiction/contradicts/inconsistent/impossible" (or equivalent) and concludes a Contradiction
claim.
Minimal inputs:
- Include the claim(s) that the reasoning trace says are in conflict.
  (a) Prefer claims explicitly referenced in the contradiction sentence(s) (e.g., Quota, Remaining, Tent/NotTent, Candidate
  claims).
  (b) If the reasoning trace's contradiction relies on an unstated or unmodeled premise, include C_MISSING.
Notes:
- This rule labels the *cause* of the contradiction (what is conflicting), not the discharge step.
- Scope must match the Contradiction claim's scope (typically BRANCH#k).
8) R_TENTATIVE_DISCHARGE  (output: Tent or NotTent in MAIN)
Essence:
- Proof-by-contradiction discharge: after Contradiction(branch=k) in BRANCH#k, the reasoning trace concludes the negation of the
tentative assumption in MAIN.
- OR, confirmation of the tentative assumption if no contradiction arose.
Inputs MUST include:
- Exactly one tentative assumption in BRANCH#k (Tent or NotTent; status="tentative")
- Exactly one Contradiction(branch=k, ...) in BRANCH#k if the output is the negation of the assumption else DO NOT include
Contradiction if confirming the assumption.
Output MUST be:
- In MAIN: the negation/confirmation of the assumption at the same cell (Tent <-> NotTent), ONLY if the reasoning trace
explicitly concludes it.
```

*Figure 14.* Possible rules for *Tents*.

```
CLAIM DESCRIPTION
Extract a faithful representation of the reasoning trace as discrete, verifiable claims. Extract EVERY claim occurrence whenever
the trace explicitly states, restates, or uses a fact matching an allowed kind | even if redundant and even if only used as
justification.
ALLOWED CLAIM KINDS + PAYLOAD SCHEMAS
- Assign:       {"r":int,"c":int,"d":int}
- Cand:         {"r":int,"c":int,"d":list[int],"polarity":"IN"|"OUT"}
- UnitMissing:  {"unit":"ROW"|"COL"|"BOX","index":int,"d":list[int]}
- UnitPlaces:   {"unit":"ROW"|"COL"|"BOX","index":int,"digit":int,"cell_r":list[int],"cell_c":list[int]}
- UnitHas:      {"unit":"ROW"|"COL"|"BOX","index":int,"digit":int}; For polarity="OUT", d is a singleton list; for
polarity="IN", d may have length >= 1.
- Contradiction: {"branch":int,"note":string}
NOTES ON CLAIM MEANINGS
Assign:
- Asserts cell (r,c) has digit d (1..9). (If 9x9 grid. For 4x4 it would be 1..4.)
- Used for given clues, placements, and hypothetical assumptions; distinguished only by metadata.
Cand:
- Candidate facts for a single cell (r,c) at that moment in the trace.
- polarity="IN": the ONLY possible digits are exactly d (exclusive).
- polarity="OUT":
    (a) digits d are eliminated (not possible); other digits may still be possible.
    (b) Represents the elimination of a SINGLE digit from a cell.
    (c) If a sentence eliminates multiple digits, extract one Cand(OUT) claim per digit.
    (d) Do not extract bundled Cand(OUT) claims with multiple digits.
- d must be non-empty, unique, ints.
- Only emit Cand(IN) if the trace explicitly asserts exclusivity (e.g., "the candidates are exactly {...}" / "can only be {...}"
/ "the only options are {...}"). Otherwise prefer Cand(OUT) or emit no Cand(IN).
UnitMissing:
- Exact list of digits missing from a unit at that moment.
- unit="ROW" => index=r, unit="COL" => index=c, unit="BOX" => index=(r//3)*3 + (c//3). (If 9x9 grid. For 4x4 it would be (r//2)*2
+ (c//2).)
- Do NOT infer updates to UnitMissing unless the trace explicitly states them.
UnitPlaces:
- For a given unit and digit, the digit can appear ONLY in the listed cells.
- cell_r and cell_c are ZIPPED coordinate pairs and must have equal non-zero length.
- Do NOT infer UnitPlaces from generic candidate eliminations unless the trace explicitly states \digit X in this unit can only
go in ...".
- cell_r/cell_c pairs must be unique and sorted lexicographically. So sort the zipped coordinate pairs by (cell_r[i], cell_c[i]).
UnitHas:
- Asserts that a unit already contains digit.
- Extract UnitHas from BOTH:
    (a) explicit inventory statements (\digits present are ...", \column has ..."),
    (b) justification statements (\cannot be 8 because 8 is in its column/row/box").
- UnitHas is verifiable and should be extracted wherever it occurs.
- If an inventory statement lists multiple digits in a unit, extract UnitHas once per listed digit.
- If a sentence says \in its row/column/box", use the referenced cell (r,c) from that same sentence to determine the unit index
only when the referenced unit is unambiguous; otherwise do NOT emit UnitHas from that sentence.
Contradiction:
- Asserts the trace declares a contradiction in a branch (with optional note).
COORDINATES AND DIGITS
- Coordinates are 0-based: top-left is (0,0).
- Digits are integers 1..9.
CLAIM OCCURRENCE RULE (CRITICAL)
- Extract a claim whenever the trace explicitly STATES OR RE-STATES OR USES a fact matching a claim kind.
    This includes:
    (a) conclusions (\so it must be ..."),
    (b) restatements/summaries (\row k is missing ..."),
    (c) premise uses / reminders (\because digit X is already in column j ...").
- Each sentence containing such an occurrence produces its own claim object.
    Do NOT deduplicate across sentences, even if kind/payload identical.
CANONICAL FORM (ALL LIST FIELDS)
- For any list[int] field (e.g., Cand.d, UnitMissing.d), sort ascending and remove duplicates.
DO NOT COMPUTE IMPLIED CONSEQUENCES
- Do NOT infer new claims that are not explicitly stated in the trace.
    Example: UnitMissing(COL,0,[5,9]) and Assign(...)=5 does NOT license adding UnitMissing(COL,0,[9])
    unless the trace explicitly states the updated missing set.
METADATA (REQUIRED ON EVERY CLAIM)
- status: "stated"|"derived"|"tentative"
- scope: "MAIN" | "BRANCH#<int>"
- source_sents: ["S#", ...] (NON-EMPTY; sentences that DIRECTLY support the claim occurrence)
- source_sents must include the sentence that contains the claim occurrence itself.
STATUS SEMANTICS
- stated: given clue/parameter explicitly presented as a given in the trace
- derived: non-hypothetical claim occurrence (placements, eliminations, unit facts, summaries)
- tentative: hypothetical/assumed inside a branch (\suppose/assume/if ...")
```

*Figure 15.* Possible claims for *Sudoku*.

*Table 15.* Puzzle size specifications by difficulty level and computational complexity.

| Puzzle | Description | Trivial | Easy | Normal | Hard | Complexity | Reference |
|---|---|---|---|---|---|---|---|
| Solo | Figure 31 | 4x4dt | 9x4dt | 9x9dt | 9x9de | NP-complete | Yato & Seta (2003) |
| Tents | Figure 32 | 4x4de | 6x6de | 8x8de | 15x15de | NP-complete | De Biasi (2012b) |
| Filling | Figure 18 | 3x3 | 9x7 | 13x9 | 17x13 | Open | – |
| Lightup | Figure 21 | 3x3d0s4 | 5x5d0s4 | 7x7d0s4 | 14x14d0s2 | NP-complete | McPhail (2005) |
| Mosaic | Figure 23 | 3x3 | 5x5 | 10x10 | 25x25 | NP-complete | de Jong (2023) |
| Keen | Figure 20 | 3x3de | 4x4de | 6x6de | 9x9dn | Open | – |
| Range | Figure 27 | 3x3 | 6x4 | 9x6 | 13x9 | Open | – |
| Pattern | Figure 25 | 2x2 | 10x10 | 15x15 | 25x25 | NP-complete | Ueda & Nagao (1996) |
| Dominosa | Figure 17 | 3x2dt | 5x4dt | 8x7db | 11x10db | Open | – |
| Galaxies | Figure 19 | 3x3dn | 5x5dn | 7x7dn | 15x15dn | NP-complete | Friedman (2002b) |
| Magnets | Figure 22 | 3x2de | 6x5de | 6x5dtS | 10x9dtS | NP-complete | Kölker (2012) |
| Net | Figure 24 | 2x2 | 4x4 | 5x5 | 11x11 | NP-complete | Král et al. (2004) |
| Pearl | Figure 26 | 5x5de | 6x6de | 8x8de | 10x10dt | NP-complete | Friedman (2002a) |
| Rect | Figure 28 | 2x1 | 5x5 | 7x7 | 15x15 | NP-complete | Takenaga et al. (2013) |
| Signpost | Figure 29 | 2x1 | 3x3 | 4x4 | 7x7 | NP-complete | Maarse (2019) |
| Singles | Figure 30 | 2x2de | 4x4de | 5x5de | 12x12de | Open | – |
| Towers | Figure 33 | 3x3de | 4x4de | 5x5de | 6x6de | NP-complete | Maarse (2019) |
| Tracks | Figure 34 | 4x4de | 8x8de | 8x8dt | 15x15dt | Open | – |
| Undead | Figure 35 | 3x3de | 4x4de | 4x4dn | 7x7dn | Open | – |
| Unequal | Figure 36 | 3x3dt | 4x4de | 4x4dk | 7x7dk | NP-complete | Maarse (2019) |
| Unruly | Figure 37 | 6x6dt | 8x6dt | 8x8dt | 14x14dt | NP-complete | De Biasi (2012a) |

```
RULE DESCRIPTION
Rule applications (apps) do NOT create claims. They only label how ONE already-extracted claim (the output) was derived in the
reasoning trace.
ALLOWED RULE IDS
- R_UNIT_HAS_STATED
- R_UNIT_MISSING_STATED
- R_ELIM_BY_UNIT_PRESENCE
- R_ELIM_BY_CONSTRAINT
- R_ONLY_REMAINING_IN_CELL
- R_ONLY_MISSING_DIGIT_IN_UNIT
- R_PATTERN_EXPERT
- R_CONTRADICTION_CAUSE
- R_CONTRADICTION_DISCHARGE
RULE APP SCHEMA
Every rule app MUST have exactly these fields:
- rule: one of the allowed rule ids
- inputs: [claim_id, ...]
- output: claim_id
- support_sents: ["S#", ...] (NON-EMPTY)
claim_id format:
- "C1", "C2"
- Special id: "C_MISSING" may appear ONLY in inputs to represent a missing premise the trace clearly relies on.
- "C_MISSING" MUST NEVER be used as output.
GLOBAL CONSTRAINTS (HARD)
G1) Wiring-only (NO NEW CLAIMS)
- output must be an existing claim_id from the provided CLAIM LIST.
- every non-missing input must be an existing claim_id from the provided CLAIM LIST.
G2) One-claim-at-a-time
- Each rule app explains exactly ONE output claim.
G3) Minimality + Faithfulness
- Inputs must match what the trace actually cites or uses for this step.
- Do NOT use claims as inputs just because they logically imply something. If the trace relies on an unstated intermediate (e.g.,
a singleton UnitMissing update), use C_MISSING rather than \nearby" claims.
- support_sents must directly justify that THIS rule produced THIS output.
- support_sents must include the sentence where the output claim is stated.
G4) Scope
- For all rules EXCEPT R_CONTRADICTION_DISCHARGE:
  all inputs and the output must have the SAME scope.
- For R_CONTRADICTION_DISCHARGE:
  inputs are from BRANCH#k and output is in MAIN.
G5) Tentative containment
- Tentative claims MUST have scope BRANCH#k (never MAIN).
- No tentative claim may influence MAIN except via R_CONTRADICTION_DISCHARGE.
G6) Rule choice must be unambiguous
- Choose the SINGLE best matching rule below; else output [].
RULE PARTITIONING (TO AVOID OVERLAP)
- R_UNIT_HAS_STATED:          UnitHas only
- R_UNIT_MISSING_STATED:      UnitMissing only
- R_ELIM_BY_UNIT_PRESENCE:    Cand(OUT) only
- R_ELIM_BY_CONSTRAINT:       Cand(OUT) only
- R_ONLY_REMAINING_IN_CELL:   Assign only
- R_ONLY_MISSING_DIGIT_IN_UNIT: Assign only
- R_PATTERN_EXPERT:           Cand(OUT) or Assign or UnitPlaces only
- R_CONTRADICTION_CAUSE:      Contradiction only
- R_CONTRADICTION_DISCHARGE:  Cand(OUT) or Assign only (in MAIN)
RULE MEANINGS (ESSENCE + MINIMAL INPUTS)
1) R_UNIT_HAS_STATED  (output: UnitHas)
Essence:
- The trace explicitly states a unit contains a digit (inventory or justification).
Minimal inputs: []
Use when:
- \digits present are ...", \column has ...", OR \because digit d is in its row/col/box".
2) R_UNIT_MISSING_STATED  (output: UnitMissing)
Essence:
- The trace explicitly states the exact missing digits of a unit.
Minimal inputs: []
Never use for:
- inferred updates after a placement unless the updated set is explicitly stated.
3) R_ELIM_BY_UNIT_PRESENCE  (output: Cand with polarity="OUT")
Essence:
- The trace eliminates digits for a cell explicitly because those digits already appear in the cell's row/col/box (\cannot be ...
because ... already appears in ...").
Minimal inputs:
- The relevant UnitHas(...) claim(s) if present; otherwise C_MISSING.
Notes:
- The output Cand(OUT).d must match exactly the eliminated digits explicitly stated.
4) R_ELIM_BY_CONSTRAINT  (output: Cand with polarity="OUT")
Essence:
- The trace eliminates digits for a cell due to an explicit constraint that is NOT phrased as direct unit presence (e.g., \by
column constraints" without naming digits present, or other explicitly cited constraint reasoning).
Minimal inputs:
- Any explicitly referenced premise claims if they exist; otherwise C_MISSING.
Tie-breaker with R_ELIM_BY_UNIT_PRESENCE:
- If the trace explicitly says \because digit X already appears in row/col/box" -> use R_ELIM_BY_UNIT_PRESENCE.
5) R_ONLY_REMAINING_IN_CELL  (output: Assign)
Essence:
- The trace concludes the digit for a cell because all other options are eliminated (\cannot be ... so it must be ..." / \only
remaining digit for the cell").
Minimal inputs:
- A Cand(OUT) elimination for that cell if present; otherwise C_MISSING.
- A UnitMissing for the relevant unit if explicitly used/stated; otherwise omit (do not force it).
Important:
- Do NOT include \implied singleton" unit facts as inputs. If needed and unstated, use C_MISSING.
6) R_ONLY_MISSING_DIGIT_IN_UNIT  (output: Assign)
Essence:
- The trace places a digit because it is the only missing digit left in a unit and the trace specifies the target cell (\only
remaining digit for row/col/box is d, placed at ...").
Minimal inputs:
- The corresponding UnitMissing(unit,index,[d]) if explicitly stated; otherwise C_MISSING.
Important:
- Do NOT treat earlier UnitMissing plus placements as an explicit singleton premise unless the trace states it.
- If the trace says \the remaining digit d is placed in cell (r,c)" without explicitly stating UnitMissing(...,[d]) (even in
previous sentences), still use this rule, but include C_MISSING as the singleton-unit premise.
7) R_PATTERN_EXPERT  (output: Cand(OUT) or Assign or UnitPlaces)
Essence:
- A named or template-identifiable advanced Sudoku technique (X-Wing, XY-Wing, Swordfish, Skyscraper, AIC/nice loop, Unique
Rectangle, ALS, etc.) is explicitly used.
Minimal inputs:
- Any claims explicitly referenced by the trace; otherwise C_MISSING.
Use only if:
- The technique name (or unambiguous template description) appears in support_sents.
8) R_CONTRADICTION_CAUSE  (output: Contradiction)
Essence:
- The trace explicitly declares a contradiction in a branch and cites statements that cannot all hold.
Minimal inputs:
- Conflicting claims if referenced; otherwise C_MISSING.
9) R_CONTRADICTION_DISCHARGE  (output: Cand(OUT) or Assign in MAIN)
Essence:
- After Contradiction in BRANCH#k, the trace concludes the negation/forced conclusion in MAIN.
Inputs MUST include:
- Exactly one tentative assumption in BRANCH#k.
- Exactly one Contradiction(branch=k, ...) in BRANCH#k.
```

*Figure 16.* Possible rules for *Sudoku*.

```
Tile the rectangle with dominoes (1×2 rectangles) so that every possible domino appears exactly once (that is, every possible
pair of numbers, including doubles).
Your task is to return a grid with the tiling. Use "L" to represent the left half of a domino and "R" to represent the right
half. Fir a vertical domino, use "T" for the top half and "B" for the bottom half. Note that it is always read from left to right
and top to bottom.
```

*Figure 17.* Puzzle description for *Dominosa*.

```
You have a grid of squares, some of which contain digits, and the rest of which are empty. Your job is to fill in digits in the
empty squares, in such a way that each connected region of squares all containing the same digit has an area equal to that digit.
('Connected region', for the purposes of this game, does not count diagonally separated squares as adjacent.)
For example, it follows that no square can contain a zero, and that two adjacent squares can not both contain a one. No region
has an area greater than 9 (because then its area would not be a single digit). Further you cannot change already filled squares.
Note that no connected regions of the same numbers can be next to eachother, since they would then be considered one connected
region. This means for instance that no squares filled with 1:s can be next to eachother.
```

*Figure 18.* Puzzle description for *Filling*.

```
You have a rectangular grid containing a number of dots. Your aim is to partition the rectangle into connected regions of
squares, in such a way that every region is 180° rotationally symmetric, and contains exactly one dot which is located at its
centre of symmetry, thus every region has to contain one dot.
To enter your solution, return the grid with all of the marked lines entered. The puzzle is complete when the marked lines on the
grid are precisely those that separate two squares belonging to different regions.
```

*Figure 19.* Puzzle description for *Galaxies*.

```
You have a square grid; each square may contain a digit from 1 to the size of the grid. The grid is divided into blocks of
varying shape and size, with arithmetic clues written in them. Your aim is to fully populate the grid with digits such that:
        Each row contains only one occurrence of each digit
        Each column contains only one occurrence of each digit
        The digits in each block can be combined to form the number stated in the clue, using the arithmetic operation given in the
        clue. That is:
                An addition clue means that the sum of the digits in the block must be the given number. For example, '15+' means the
                contents of the block adds up to fifteen.
                A multiplication clue (e.g. '60×'), similarly, means that the product of the digits in the block must be the given
                number.
                A subtraction clue will always be written in a block of size two, and it means that one of the digits in the block is
                greater than the other by the given amount. For example, '2-' means that one of the digits in the block is 2 more than
                the other, or equivalently that one digit minus the other one is 2. The two digits could be either way round, though.
                A division clue (e.g. '3÷'), similarly, is always in a block of size two and means that one digit divided by the other is
                equal to the given amount.
        Note that a block may contain the same digit more than once (provided the identical ones are not in the same row and column).
Return the grid with all of the digits entered. The puzzle is complete when the grid is fully populated and all of the arithmetic
clues are satisfied.
```

*Figure 20.* Puzzle description for *Keen*.

```
You have a grid of squares. Some are filled in black; some of the black squares are numbered. Your aim is to 'light up' all the
empty squares by placing light bulbs in some of them.
Each light bulb illuminates the square it is on, plus all squares in line with it horizontally or vertically unless a black
square is blocking the way.
To win the game, you must satisfy the following conditions:
        All non-black squares are lit.
        No light is lit by another light.
        All numbered black squares have exactly that number of lights adjacent to them (in the four squares above, below, left, and
        right).
Non-numbered black squares may have any number of lights adjacent to them.
Make sure to return the whole grid with the light bulbs marked as \"L\".
```

*Figure 21.* Puzzle description for *Lightup*.

```
A rectangular grid has been filled with a mixture of magnets (that is, dominoes with one positive end and one negative end) and
blank dominoes (that is, dominoes with two neutral poles). These dominoes are initially only seen in silhouette. Around the grid
are placed a number of clues indicating the number of positive or negative poles contained in certain columns and rows.
Your aim is to correctly place the magnets and blank dominoes such that all the clues are satisfied, with the additional
constraint that no two similar magnetic poles may be orthogonally adjacent (since they repel). Neutral poles do not repel, and
can be adjacent to any other pole.
Make sure to use 1s to represent positive poles, 0s to represent neutral poles, and -1s to represent negative poles. The grid
should be filled with 1s, 0s, and -1s only, representing the dominoes. Do not return the clues or any other information.
Note that all poles must be magnets placed in the existing domio places, thus no domino can have to negative poles.
```

*Figure 22.* Puzzle description for *Magnets*.

```
You are given a grid of squares, which you must colour either black or white.
Some squares contain clue numbers. Each clue tells you the number of black squares in the 3×3 region surrounding the clue {
including the clue square itself.
You have to return a filled grid with all of the squares coloured. Black is represented by 1, white by 2.
```

*Figure 23.* Puzzle description for *Mosaic*.

The computer prepares a network by connecting up the centres of squares in a grid, and then shuffles the network by rotating every tile randomly. Your job is to rotate it all back into place. The successful solution will be an entirely connected network, with no closed loops, i.e. all sinks are connected to the source.
Return the grid with the degree of rotation of each tile (0, 90, 180, or 270 degrees) in the grid, such that the source is connected to all of the sinks. Note that the number which should be returned for each cell is the degree of the tile, not how many degrees the tile has to be rotated.

*Figure 24.* Puzzle description for *Net*.

You have a grid of squares, which must all be filled in either black or white. Beside each row of the grid are listed, in order, the lengths of the runs of black squares on that row; above each column are listed, in order, the lengths of the runs of black squares in that column. Your aim is to fill in the entire grid black or white.
You have to return the entire grid filled out. Do not return the row and column clues, ignores those when returning the grid, thus the grid will be one column and one row shorter than the input grid with the clues.

*Figure 25.* Puzzle description for *Pattern*.

You have a grid of squares. Your job is to draw lines between the centres of horizontally or vertically adjacent squares, so that the lines form a single closed loop. In the resulting grid, some of the squares that the loop passes through will contain corners, and some will be straight horizontal or vertical lines. (And some squares can be completely empty { the loop doesn't have to pass through every square, fill these squares with −1.)
Some of the squares contain black and white circles, which are clues that the loop must satisfy.
A black circle in a square indicates that that square is a corner, but neither of the squares adjacent to it in the loop is also a corner.
A white circle indicates that the square is a straight edge, but at least one of the squares adjacent to it in the loop is a corner.
Return the grid with all of the lines entered as numbers in consequtive numbers, starting with 1 as the first number (then 2 would be the next connected square forming a lines with 1) and let unmarked ones be −1. Note that all numbers have to be consequtive and either above/below or left/right, not diagonally, and the 1 has to be next to the highest number in order to form a closed loop. The puzzle is complete when the lines created by the numbers on the grid form a single closed loop, and the loop satisfies all of the clues.

*Figure 26.* Puzzle description for *Pearl*.

You have a grid of squares; some squares contain numbers. Your job is to colour some of the squares black, such that several criteria are satisfied:
    no square with a number is coloured black.
    no two black squares are adjacent (horizontally or vertically).
    for any two white squares, there is a path between them using only white squares.
    for each square with a number, that number denotes the total number of white squares reachable from that square going in a straight line in any horizontal or vertical direction until hitting a wall or a black square; the square with the number is included in the total (once).
For instance, a square containing the number one must have four black squares as its neighbours by the last criterion; but then it's impossible for it to be connected to any outside white square, which violates the second to last criterion. So no square will contain the number one.
Return the grid with all of the black squares entered as −1, let unmarked ones be 0.

*Figure 27.* Puzzle description for *Range*.

You have a grid of squares, with numbers written in some (but not all) of the squares. Your task is to subdivide the grid into rectangles of various sizes, such that (a) every rectangle contains exactly one numbered square, and (b) the area of each rectangle is equal to the number written in its numbered square.
Return the entire grid filled out with each grid having a unique number. Mark the cells of each rectangle with the same number.
Do not return the numbers in the grid, only the rectangles in the grid.

*Figure 28.* Puzzle description for *Rectangles*.

You have a grid of squares; each square (except the last one) contains an arrow, and some squares also contain numbers. Your job is to connect the squares to form a continuous list of numbers starting at 1 and linked in the direction of the arrows { so the arrow inside the square with the number 1 will point to the square containing the number 2, which will point to the square containing the number 3, etc. Each square can be any distance away from the previous one, as long as it is somewhere in the direction of the arrow.
Note that all cells have to be used, and the path must be continuous.
By convention the first and last numbers are shown; one or more interim numbers may also appear at the beginning.
Return the grid with all of the numbers entered as integers. The puzzle is complete when all of the numbers are entered and connected in the correct order.

*Figure 29.* Puzzle description for *Signpost*.

You have a grid of white squares, all of which contain numbers. Your task is to colour some of the squares black (removing the number) so as to satisfy all of the following conditions:
    No number occurs more than once in any row or column.
    No black square is horizontally or vertically adjacent to any other black square.
    The remaining white squares must all form one contiguous region (connected by edges, not just touching at corners).
Return the grid with all of the black squares filled in as −1.

*Figure 30.* Puzzle description for *Singles*.

```
You have a square grid, which is divided into as many equally sized sub-blocks as the grid has rows. Each square must be filled
in with a digit from 1 to the size of the grid, in such a way that:
1. every row contains only one occurrence of each digit
2. every column contains only one occurrence of each digit
3. every block contains only one occurrence of each digit.
You are given some of the numbers as clues; your aim is to place the rest of the numbers correctly.
```

*Figure 31.* Puzzle description for *Solo* (also known as *Sudoku*).

```
You have a grid of squares, some of which contain trees. Your aim is to place tents in some of the remaining squares, in such a
way that the following conditions are met:
There are exactly as many tents as trees.
The tents and trees can be matched up in such a way that each tent is directly adjacent (horizontally or vertically, but not
diagonally) to its own tree. However, a tent may be adjacent to other trees as well as its own.
No two tents are adjacent horizontally, vertically or diagonally.
The number of tents in each row, and in each column, matches the numbers given in the row or column constraints.
Grass indicates that there cannot be a tent in that position.
```

*Figure 32.* Puzzle description for *Tents*.

```
You have a square grid. On each square of the grid you can build a tower, with its height ranging from 1 to the size of the grid.
Around the edge of the grid are some numeric clues.
Your task is to build a tower on every square, in such a way that:
    Each row contains every possible height of tower once
    Each column contains every possible height of tower once
    Each numeric clue describes the number of towers that can be seen if you look into the square from that direction, assuming
    that shorter towers are hidden behind taller ones. For example, in a 5×5 grid, a clue marked '5' indicates that the five
    tower heights must appear in increasing order (otherwise you would not be able to see all five towers), whereas a clue marked
    '1' indicates that the tallest tower (the one marked 5) must come first.
In harder or larger puzzles, some towers will be specified for you as well as the clues round the edge, and some edge clues may
be missing.
Return the grid with all of the towers entered as integers. The puzzle is complete when all of the towers are entered and the
clues are satisfied. You should not return the clues in the output grid.
```

*Figure 33.* Puzzle description for *Towers*.

```
You are given a grid of squares, some of which are filled with train tracks. You need to complete the track from A to B so that
the rows and columns contain the same number of track segments as are indicated in the clues to the top and right of the grid.
There are only straight and 90 degree curved rails, and the track may not cross itself.
Tracks was contributed to this collection by James Harvey.
 Left-click on an edge between two squares adds a track segment between the two squares. Right-clicking on an edge adds a
 cross on the edge, indicating no track is possible there.
Left-clicking in a square adds a colour indicator showing that you know the square must contain a track, even if you don't know
which edges it crosses yet. Right-clicking in a square adds a cross indicating it contains no track segment.
Left- or right-dragging between squares allows you to lay a straight line of is-track or is-not-track indicators, useful for
filling in rows or columns to match the clue.
```

*Figure 34.* Puzzle description for *Tracks*.

```
You are given a grid of squares, some of which contain diagonal mirrors. Your goal is to fill them with the given amount of each
type monster. Every square which is not a mirror must be filled with one of three types of undead monster: a ghost, a vampire, or
a zombie.
Vampires can be seen directly, but are invisible when reflected in mirrors. Ghosts are the opposite way round: they can be seen
in mirrors, but are invisible when looked at directly. Zombies are visible by any means.
You are also told the total number of each type of monster in the grid. Also around the edge of the grid are written numbers,
which indicate how many monsters can be seen if you look into the grid along a row or column starting from that position. (Note
the following: The diagonal mirrors are reflective on both sides. If your reflected line of sight crosses the same monster more
than once, the number will count it each time it is visible, not just once.)
You need to return the grid filled with the monsters, such that the number of each type of monster is correct, and the numbers
around the edge of the grid are satisfied. You should not return the numbers around the edge of the grid, just the grid itself.
Make sure to place the monesters in empty squares, where the following symbol indicating the type of monster: "G" for a ghost,
"V" for a vampire, or "Z" for a zombie.
```

*Figure 35.* Puzzle description for *Undead*.

```
 You have a square grid; each square may contain a digit from 1 to the size of the grid, and some squares have clue signs between
 them. Your aim is to fully populate the grid with numbers such that:
    Each row contains only one occurrence of each digit
    Each column contains only one occurrence of each digit
    All the clue signs are satisfied.
Make sure to return the whole grid with the correct numbers. Do not include the additional conditions and constraint. Remember
that it is not possible to overwrite the prefilled non zero numbers.
```

*Figure 36.* Puzzle description for *Unequal*.

```
You are given a grid of squares, which you must colour either black or white. Some squares are provided as clues; the rest are
left for you to fill in. Each row and column must contain the same number of black and white squares, and no row or column may
contain three consecutive squares of the same colour.
Make sure to return the whole grid with the colors entered.
```

*Figure 37.* Puzzle description for *Unruly*.

