# OpenReview forum: "Reasoning Structure of Large Language Models"
_ICML.cc/2026/Conference — ICML 2026 regular_

### Official Review · Reviewer_2LnB · 2026-03-12

**Soundness:** 3
**Presentation:** 2
**Significance:** 3
**Originality:** 3
**Overall Recommendation:** 4
**Confidence:** 3

**Summary:**

Overall, the authors outline the central challenge of evaluating large reasoning models beyond simplistic scalar metrics like final-answer accuracy or token counts. To address this, the paper proposes an automated evaluation framework based on 2D grid logic puzzles. By converting unstructured chain-of-thought texts into reasoning graphs of claims and dependencies, and introducing a metric based on structural information entropy, the authors attempt to quantitatively characterize the focus and redundancy of the intermediate reasoning flow.

**Compliance With Llm Reviewing Policy:**

Affirmed.

**Final Justification:**

My concerns are addressed and have raised the score from 3 to 4.

**Key Questions For Authors:**

1. How are edges “verified”? Beyond LLM rule attribution, do you implement rule checkers that confirm a proposed derivation is valid under puzzle semantics, or human annotation for a subset? What is the measured precision/recall of edge extraction?
2. How sensitive are Hstr and η to (a) the initial distribution over source nodes and (b) transition weighting? Could you report a sensitivity analysis across a few plausible alternatives?
3. Many state-dependent claims are not verified mid-trace (Appendix B). What fraction of claims per puzzle/difficulty remain unverifiable, and how might this bias claim-accuracy and its correlation with η?

**Limitations:**

yes

**Strengths And Weaknesses:**

Strengths

Shifting the evaluation paradigm from static final answers to the dynamic topological structure of the reasoning process is a highly relevant research direction. Utilizing an executable puzzle environment with deterministic state transitions to perform objective, claim-level verification provides a much more rigorous testbed for diagnosing invalid exploration in long-horizon reasoning compared to standard static NLP datasets.

Weaknesses
1. Heavy reliance on LLMs for claim and rule extraction introduces potential bias and circularity; while node-level verification is supported, edge correctness appears to rely largely on LLM inference with limited formal checking.
2. The Markov-chain model assumes uniform initial mass over source nodes and unweighted transitions; no sensitivity analysis is provided for these modeling choices or alternatives (e.g., time-/text-weighted flows).
3. η depends on the minimal claim set C* and extracted graph size |V|. For partially solved or unsolved instances, applicability and interpretation are less clear, and the paper primarily evaluates η on solved cases.
4. Limited ablations on the extraction pipeline (e.g., deterministic-only vs. LLM-only vs. hybrid; swapping extractor models) to assess robustness and generalization.
5. Lack of human or programmatic validation of edge derivations beyond LLM rule attribution; the claim “both nodes and edges can be checked” would benefit from explicit edge verifiers or inter-annotator agreement studies.

---

> ### Author Rebuttal · Authors · 2026-03-31
>
> We thank the reviewer for the careful evaluation and address each point below.
>
> ### LLM reliance for extraction and edge correctness
>
> > Heavy reliance on LLMs for extraction introduces potential bias [...] edge correctness relies largely on LLM inference with limited formal checking.
>
> We appreciate this concern. LLM reliance is a real limitation, though it is also what makes structural analysis feasible at scale for the trace lengths we analyze.
>
> To quantify edge quality, we manually evaluated 200 randomly sampled rule applications: 151 (75.5%) were fully correct. Most errors involved a missing premise rather than spurious edges (75.5% is conservative, as a single missing premise counts as a full error). The framework remains robust overall: claim verification is deterministic, the extractor ablation across six LLMs shows consistent trends (see Reviewer fcqF), and perturbation analysis shows $\eta$ CV below 5% for 6x6+ graphs (see Reviewer fcqF).
>
> ### Markov chain assumptions and sensitivity
>
> > Uniform initial mass and unweighted transitions, no sensitivity analysis provided. [...] How sensitive are $H_\text{str}$ and $\eta$ to (a) the initial distribution and (b) transition weighting?
>
> We tested 7 alternatives against the default (uniform $\pi$, uniform $P$, mean $\eta = 0.158$): two initial distributions, four transition weightings, and one combined variant.
>
> | Method | $\rho(\eta)$ | $\vert\Delta\eta\vert$ | Mean $\eta$ |
> |---|---|---|---|
> | Text-proximity $P$ | 0.993 | 0.009 | 0.149 |
> | Exp-decay $P$ | 0.968 | 0.018 | 0.145 |
> | Inverse-fan-in $P$ | 0.964 | 0.028 | 0.130 |
> | Subtree-weighted $P$ | 0.861 | 0.062 | 0.218 |
> | Recency $\pi$ | 0.850 | 0.118 | 0.276 |
> | Recency $\pi$ + Text-prox $P$ | 0.847 | 0.109 | 0.267 |
> | Degree $\pi$ | 0.778 | 0.074 | 0.228 |
>
> All transition-weighting variants preserve near-perfect instance ranking ($\rho(\eta) \geq 0.86$, $\vert\Delta\eta\vert \leq 0.062$). Initial-distribution variants show more shift but maintain $\rho(\eta) \geq 0.78$. $H_\text{str}$ is even more stable ($\rho \geq 0.994$ across all 7 variants). The variation in $\eta$ stems primarily from the normalization rather than the underlying entropy, confirming the metric captures intrinsic graph structure.
>
> ### $\eta$ for partially solved/unsolved instances
>
> > $\eta$ depends on the minimal claim set $C^*$ and extracted graph size $|V|$. For partially solved or unsolved instances, applicability and interpretation are less clear.
>
> C∗ is still well-defined for unsolved instances (it is the puzzle solution). We compute $\eta$ using whatever correct solution claims appear in the trace. Incorrect traces show 57% lower efficiency, 12% lower claim accuracy, and first errors 53% earlier (see our response to Reviewer z5Q6)
>
> ### Limited extraction pipeline ablations
>
> > Limited ablations on the extraction pipeline (e.g., deterministic-only vs. LLM-only vs. hybrid; swapping extractor models) to assess robustness and generalization.
>
> See our response to Reviewer fcqF (Extractor ablation) for the full ablation across six extraction LLMs. Directional trends are preserved and no self-bias is observed.
>
> ### Edge validation and precision/recall
>
> > "Both nodes and edges can be checked" would benefit from explicit edge verifiers [...] What is the measured precision/recall of edge extraction?
>
> We agree the wording in Section 2.3 was misleading and have revised it. Nodes are verified deterministically against the puzzle environment (Section 3.4). Edges rely on LLM rule attribution with puzzle-specific templates (Appendix D.4) that constrain the space of valid derivations. We do not currently have programmatic edge checkers. Our human evaluation of 200 randomly sampled rule applications (see LLM reliance response above) found 75.5% fully correct under a strict criterion where a single missing premise counts as a full error. Most errors were incomplete derivations rather than spurious edges.
>
> ### Fraction of unverifiable claims and bias on η
>
> > What fraction of claims per puzzle/difficulty remain unverifiable, and how might this bias claim-accuracy and its correlation with $\eta$?
>
> These are state-dependent claims referencing the current board state. We exclude them from verification to avoid cascading grading errors. Fractions per model and size (mean ± std):
>
> | Model | 4 | 5 | 6 | 7 |
> |---|---|---|---|---|
> | Qwen3 | 0.00±0.00 | 0.13±0.16 | 0.11±0.11 | 0.07±0.08 |
> | DeepSeek | 0.03±0.06 | 0.06±0.10 | 0.07±0.07 | 0.14±0.14 |
> | Kimi | 0.00±0.00 | 0.13±0.19 | 0.08±0.09 | 0.14±0.17 |
>
> Fractions are modest (under 15%). Errors in these claims typically propagate to later verifiable claims, so the practical effect is a slightly delayed first-error detection. We retain these claims in the graph to preserve reasoning topology. The main potential bias is a slight attenuation of the $\eta$-accuracy correlation.

---

> > ### Author Rebuttal · Reviewer_2LnB · 2026-04-03
> >
> > I thank the author for their detailed reviews. My concerns are addressed and have raised the score from 3 to 4.

---

### Official Review · Reviewer_fcqF · 2026-03-13

**Soundness:** 4
**Presentation:** 4
**Significance:** 3
**Originality:** 3
**Overall Recommendation:** 5
**Confidence:** 4

**Summary:**

This paper proposes a framework for analyzing the structure of reasoning in large reasoning models (LRMs), moving beyond standard metrics like final-answer accuracy and token count. The authors introduce: (1) a scalable benchmark of 21 grid-based logic puzzles at four difficulty levels, (2) an automated pipeline that converts free-form textual reasoning traces into verifiable directed acyclic graphs (DAGs) of atomic claims and deductive dependencies, and (3) a reasoning-flow efficiency metric $\eta$, grounded in absorbing Markov chain theory, that measures how concentrated a model's logical flow is relative to the minimal claim set needed for the solution. Experiments on four frontier/open-source models (GPT-5, Qwen3 235B, DeepSeek V3.2, Kimi K2) reveal that token count is a poor proxy for reasoning quality, that extra tokens mainly translate into verification overhead rather than improved structure, and that η captures meaningful differences between focused and diffuse reasoning even when accuracy and token counts are similar.

**Compliance With Llm Reviewing Policy:**

Affirmed.

**Final Justification:**

The authors addresses nearly all concerns with robustness of their metric against model-dependent factors, therefore I am raising my evaluation to 5: accept.

**Key Questions For Authors:**

- The paper uses temperature T = 1 for all models. Since reasoning performance is known to be sensitive to temperature, was this choice validated? Lower temperatures might yield more focused traces, potentially affecting $\eta$.
- How sensitive is $\eta$ to the granularity of claim extraction? If the extraction pipeline misses claims or hallucinates edges, how does this propagate to $\eta$? A synthetic experiment would be informative.
- Have you tested the extraction pipeline with a different LLM (e.g., Claude, Llama) in place of GPT-5.2 for claim extraction? A positive result here, showing robustness to extractor choice, would significantly strengthen the claims.

**Limitations:**

Yes

**Strengths And Weaknesses:**

Strengths:

The core observation that identical accuracy and token counts can mask fundamentally different reasoning behaviors is important and well-argued. The paper makes a convincing case that the field needs structural tools for understanding how models reason, not just whether they get the right answer.

The reasoning-flow efficiency $\eta$ is theoretically grounded via absorbing Markov chains and structural entropy. Figure 5 provides a rich and informative set of scatter plots with statistical tests. The metric has desirable properties: it is uncorrelated with raw token count, positively correlated with solution-supporting graph fraction, and positively correlated with claim accuracy, suggesting it captures something genuinely meaningful about reasoning quality. The finding that verification overhead grows strongly with token count while $\eta$ does not is a key insight.

The suite of 21 puzzles spanning diverse constraint types with four controlled difficulty levels is a significant contribution. Building on Simon Tatham's puzzle collection with an executable environment that enables intermediate claim verification is a strong design choice.

Weaknesses:

The graph extraction pipeline uses GPT-5.2 for claim extraction/screening and GPT-5-mini for rule extraction. This creates a significant methodological concern: the structural properties being measured are filtered through the capabilities and biases of another LLM, and the paper never tests an alternative extractor. No ablation is provided swapping GPT-5.2 for a different extraction model to verify that the resulting graphs and $\eta$ values are robust to extractor choice. The authors' defense in Section 6, that bias is reduced by "separating roles" between GPT-5.2 (claim extraction) and GPT-5-mini (rule extraction), is a design argument, not an empirical validation, and not based on existing literature. The stability analysis in Table 2, while showing reasonable repeatability, reveals that extraction reliability varies across source models: Jaccard overlap is 0.98 for Qwen3 traces but only 0.79 for DeepSeek traces. This suggests the extraction pipeline has different reliability for different source models, which is concerning given that the paper draws structural comparisons across these same models. Without an extractor-model ablation, it is difficult to disentangle genuine differences in reasoning structure from differential extraction fidelity.

The paper does not calculate $\eta$ on failed traces. This could be a missed opportunity for analysis. Computing $eta$ on failed traces could reveal whether low efficiency predicts failure, whether different failure modes (e.g., early errors vs. diffuse exploration vs. circular reasoning) produce distinct eta signatures, and how $\eta$ distributes differently for solved vs. unsolved instances of the same puzzle. The hardest difficulty level, where all open-source models achieve 0% accuracy, would be particularly informative but goes entirely unanalyzed structurally.

The paper never reports the number of solved instances underlying the structural analysis in Figures 5 and 6 or how they were sampled. Could be heavily skewed toward easier difficulty levels: nearly all solved instances come from Trivial and Easy. The paper should explicitly report the sample sizes and difficulty distribution underlying each figure, and discuss how any difficulty imbalance affects the generalizability of the observed correlations.

---

> ### Author Rebuttal · Authors · 2026-03-31
>
> We thank the reviewer for the detailed and rigorous feedback. We address each concern below.
>
> ### Extractor ablation and robustness
>
> > No ablation is provided swapping GPT-5.2 for a different extraction model [...] extraction reliability varies across source models: Jaccard 0.98 for Qwen3 but only 0.79 for DeepSeek. [...] Have you tested with a different LLM (e.g., Claude, Llama)?
>
> We repeated extraction using five additional LLMs on the same traces (means ± std over 3 runs, excluding deterministically extracted solution claims, so Jaccard values are conservative lower bounds):
>
> | Extractor | Source | Jaccard | $H_\text{str}$ | $\eta$ |
> |---|---|---|---|---|
> | Main | Kimi | 0.89±0.08 | 3.38±0.09 | 0.15±0.06 |
> | Main | DeepSeek | 0.79±0.07 | 3.39±0.07 | 0.14±0.06 |
> | Main | Qwen3 | 0.98±0.01 | 3.66±0.06 | 0.15±0.04 |
> | GPT-5.4 | Kimi | 0.81±0.07 | 3.47±0.25 | 0.21±0.08 |
> | GPT-5.4 | DeepSeek | 0.98±0.01 | 3.40±0.06 | 0.13±0.06 |
> | GPT-5.4 | Qwen3 | 0.92±0.04 | 3.43±0.22 | 0.21±0.05 |
> | DeepSeek | Kimi | 0.88±0.06 | 3.45±0.22 | 0.20±0.08 |
> | DeepSeek | DeepSeek | 0.98±0.02 | 3.40±0.07 | 0.11±0.04 |
> | DeepSeek | Qwen3 | 0.91±0.01 | 3.44±0.20 | 0.21±0.05 |
> | Qwen3 | Kimi | 0.92±0.03 | 3.32±0.03 | 0.20±0.10 |
> | Qwen3 | DeepSeek | 0.90±0.03 | 3.41±0.08 | 0.10±0.03 |
> | Qwen3 | Qwen3 | 0.92±0.01 | 3.31±0.03 | 0.29±0.05 |
> | Kimi | Kimi | 0.67±0.06 | 3.42±0.22 | 0.22±0.10 |
> | Kimi | DeepSeek | 0.90±0.07 | 3.37±0.05 | 0.12±0.03 |
> | Kimi | Qwen3 | 0.71±0.09 | 3.46±0.22 | 0.20±0.05 |
> | GPT-OSS | Kimi | 0.83±0.07 | 3.40±0.21 | 0.22±0.09 |
> | GPT-OSS | DeepSeek | 0.65±0.05 | 3.38±0.20 | 0.21±0.09 |
> | GPT-OSS | Qwen3 | 0.77±0.10 | 3.33±0.09 | 0.22±0.04 |
>
> Directional trends are consistent: DeepSeek yields the lowest $\eta$ in 6/6 extractors ($(1/3)^6 \approx 0.1\%$ under random ordering). No self-bias is observed. $H_\text{str}$ has a CV of only 1.9% across extractors.
>
> ### $\eta$ on failed traces and unanalyzed difficulty levels
>
> > The paper does not calculate $\eta$ on failed traces [...] The hardest difficulty level, where all open-source models achieve 0% accuracy, goes entirely unanalyzed structurally.
>
> We agree this is a valuable extension. See our response to Reviewer z5Q6.
>
> ### Sample sizes and difficulty distribution
>
> > The paper never reports the number of solved instances underlying the structural analysis in Figures 5 and 6 or how they were sampled. Could be heavily skewed toward easier difficulty levels.
>
> Figures 5-6 use solved instances only (3 traces × 3 instances per size per model):
>
> | Model | 4 | 5 | 6 | 7 | Total |
> |---|---|---|---|---|---|
> | DeepSeek | 6 | 3 | 5 | 6 | 20 |
> | Kimi | 9 | 8 | 9 | 9 | 35 |
> | Qwen3 | 9 | 9 | 7 | 5 | 30 |
> | **Total** | 24 | 20 | 21 | 20 | 85 |
>
> The distribution is balanced across sizes. When stratifying by puzzle size (see our response to Reviewer ErHL), the token-$\eta$ relationship remains near zero at every size, confirming the pooled findings are not driven by difficulty imbalance.
>
> ### Temperature $T=1$
>
> > The paper uses temperature T = 1 for all models. Since reasoning performance is known to be sensitive to temperature, was this choice validated?
>
> Our framework is temperature-agnostic. We chose $T=1$ for consistency, noting that low temperatures cause repetitive loops in reasoning models (Pipis et al., 2025) and that decoding strategy affects model rankings (Song et al., NAACL 2025). Since we analyze reasoning structure rather than benchmark capabilities, the temperature choice does not affect the methodology.
>
> - Pipis, C. et al. "Wait, Wait, Wait... Why Do Reasoning Models Loop?" arXiv:2512.12895, 2025.
> - Song, Y. et al. "The Good, The Bad, and The Greedy: Evaluation of LLMs Should Not Ignore Non-Determinism." NAACL 2025.
>
> ### Sensitivity of $\eta$ to extraction granularity
>
> > How sensitive is $\eta$ to the granularity of claim extraction? If the extraction pipeline misses claims or hallucinates edges, how does this propagate to $\eta$?
>
> We apply three perturbation types per graph: drop one node, drop one edge, or add 1-4 random edges. $\eta$ shows comparable sensitivity to simpler metrics (depth, diameter), and sensitivity diminishes with graph size ($\eta$ CV by puzzle size):
>
> | Perturbation | 4x4 | 5x5 | 6x6 | 7x7 |
> |---|---|---|---|---|
> | Node removal | 0.130 | 0.079 | 0.051 | 0.037 |
> | Edge removal | 0.115 | 0.060 | 0.043 | 0.035 |
> | Edge add (1) | 0.165 | 0.074 | 0.051 | 0.038 |
> | Edge add (2) | 0.260 | 0.107 | 0.075 | 0.052 |
> | Edge add (3) | 0.300 | 0.134 | 0.091 | 0.071 |
> | Edge add (4) | 0.340 | 0.167 | 0.118 | 0.081 |
>
> For 6x6 and above (where most of our analysis operates), single-perturbation CV is below 5%.

---

> > ### Author Rebuttal · Reviewer_fcqF · 2026-04-02
> >
> > Thank you for the rebuttal and the additional ablation results you provided. I believe the metric you introduced is robust against model-dependent factors and provides valuable information that traditional and simpler structural metrics do not provide. I have updated my score accordingly. I would love to see the results of the larger-scale matched analysis for unsolved instances when that's available.

---

### Official Review · Reviewer_ErHL · 2026-03-13

**Soundness:** 3
**Presentation:** 3
**Significance:** 3
**Originality:** 3
**Overall Recommendation:** 4
**Confidence:** 3

**Summary:**

This paper studies the structure of reasoning processes in large language models (LLMs), arguing that commonly used evaluation metrics such as final answer accuracy and token count are insufficient to characterize how models reason. The authors observe that two reasoning traces may achieve the same answer accuracy with similar token budgets while exhibiting very different reasoning behaviors.

To address this limitation, the paper proposes representing chain-of-thought reasoning traces as reasoning graphs, where nodes correspond to atomic claims and edges represent deductive dependencies. Based on this structured representation, the authors introduce a metric called reasoning-flow efficiency (η), which measures how concentrated the logical flow of reasoning is relative to the minimal claim set required for the solution.

The authors evaluate their approach on a benchmark consisting of 21 logic puzzles with multiple difficulty levels. Reasoning traces from several reasoning models (e.g., GPT-5, Qwen3-235B, DeepSeek-V3.2, and Kimi-K2) are converted into reasoning graphs, and the proposed metric is used to analyze relationships between reasoning efficiency, token usage, claim accuracy, and error depth. The results suggest that token count alone is not strongly correlated with the proposed efficiency metric and that reasoning structure can reveal differences between reasoning traces that are not captured by traditional evaluation metrics.

**Compliance With Llm Reviewing Policy:**

Affirmed.

**Key Questions For Authors:**

Questions for the Authors
	1.	On the relationship between token usage and reasoning quality.
The paper argues that increasing token budgets does not necessarily improve reasoning performance and that token count is not a reliable proxy for reasoning quality. However, the analysis in Table 1 compares models across different difficulty levels and model families simultaneously. Could the authors provide additional analysis of the relationship between token usage and reasoning quality within the same model and the same difficulty level? For example, is reasoning quality positively correlated with token usage within the same task setting?
	2.	On the interpretation of traditional metrics.
The paper suggests that accuracy and token count are insufficient to distinguish reasoning behaviors. However, in Table 1, traditional metrics already capture some meaningful differences between models (e.g., Kimi-K2 uses more tokens but achieves lower accuracy than GPT-5). Could the authors clarify in which scenarios the proposed structural metrics lead to substantially different conclusions compared to traditional evaluation metrics?
	3.	On the applicability of reasoning graphs beyond puzzle environments.
The proposed framework relies on the ability to extract and verify atomic claims using an executable puzzle environment. How do the authors envision extending this approach to more general reasoning tasks where such verifiable intermediate states may not exist?

**Limitations:**

Yes

**Strengths And Weaknesses:**

Strengths

1. Interesting perspective on reasoning evaluation.
The paper proposes analyzing LLM reasoning from a structural perspective rather than relying solely on outcome metrics such as accuracy or token count. Representing reasoning traces as graphs of claims and dependencies provides a potentially useful way to study reasoning behavior.

2. Structured representation of reasoning traces.
Converting chain-of-thought reasoning into reasoning graphs enables analysis of reasoning topology, such as solution-supporting paths, verification overhead, and error propagation. This representation could provide useful insights into reasoning patterns and failure modes.

3. A new metric for reasoning efficiency.
The proposed reasoning-flow efficiency metric attempts to quantify how focused the reasoning process is on the minimal set of claims required for the solution. Such a metric could potentially help analyze phenomena such as overthinking or redundant reasoning steps.

Weaknesses

1. Limited generality of the proposed framework.
The proposed method relies on puzzle environments with well-defined rules and verifiable intermediate states, which allows the extraction and verification of claims in the reasoning graph. However, it is unclear how easily this framework can be extended to more general reasoning tasks, such as open-domain question answering or natural language reasoning, where claims and dependencies may not be explicitly verifiable.
2. Benchmark scope is relatively narrow.
The experimental evaluation focuses exclusively on logic puzzles with deterministic rules. While such tasks provide a controlled environment for studying reasoning behavior, they may not fully represent the diversity of reasoning tasks encountered in real-world LLM applications.

---

> ### Author Rebuttal · Authors · 2026-03-31
>
> We thank the reviewer for the thoughtful comments, especially the question on within-model token–quality correlations, which prompted a cleaner analysis.
>
> ### Limited generality beyond puzzle environments
>
> > The proposed method relies on puzzle environments with well-defined rules [...] it is unclear how easily this framework can be extended to more general reasoning tasks.
>
> Our paper focuses on establishing the framework and demonstrating its value in a controlled setting where ground-truth verification is possible. Puzzles are chosen precisely because they allow deterministic claim verification (Section 3.4), which is necessary to validate that $\eta$ measures something meaningful. Extending to open-domain reasoning would require replacing deterministic verification with approximate methods (e.g., entailment models or LLM judges), which introduces its own challenges. Establishing the methodology in a fully verifiable setting allows us to validate that $\eta$ measures something meaningful before moving to noisier domains. We expand the discussion of extensibility in the revised Section 6.
>
> ### Narrow benchmark scope
>
> > The experimental evaluation focuses exclusively on logic puzzles with deterministic rules [...] may not represent the diversity of real-world reasoning tasks.
>
> We agree that grid puzzles are a specific domain. However, our 21 puzzle families span diverse constraint types (placement, connectivity, counting, Latin-square constraints, see Figure 2) and four difficulty levels. The environment also scales to larger puzzle sizes as models improve. The key insight, that token count is a poor proxy for reasoning quality, and the graph-based methodology apply wherever intermediate reasoning steps can be extracted and verified.
>
> ### Token usage vs quality within same model and difficulty
>
> > Could the authors provide additional analysis of the relationship between token usage and reasoning quality within the same model and the same difficulty level?
>
> We stratified correlations by model and by puzzle size. Token-η correlations are near zero at every level, while token-accuracy correlations are consistently negative and strengthen with size:
>
> | Model/Size | Tok vs. $\eta$ | Tok vs. Acc |
> |---|---|---|
> | DeepSeek | −0.399 | −0.756 |
> | Kimi | +0.176 | −0.388 |
> | Qwen3 | −0.397 | −0.770 |
> | 4×4 | +0.020 | −0.324 |
> | 5×5 | −0.028 | −0.604 |
> | 6×6 | −0.032 | −0.532 |
> | 7×7 | −0.040 | −0.771 |
>
> The negative token-accuracy relationship is driven by difficulty. Harder puzzles produce wider, more diffuse graphs with more tokens and lower accuracy (tokens vs. width $r = +0.652$, width vs. accuracy $r = -0.582$). This is consistent with recent findings that token count proxies for problem difficulty, not reasoning quality (Muennighoff et al., 2025, Shojaee et al., 2025). $\eta$ remains uncorrelated with tokens at every stratification level yet positively associated with accuracy ($r = +0.334$), confirming it captures reasoning focus rather than difficulty or verbosity.
>
> ### When do structural metrics lead to different conclusions?
>
> > In Table 1, traditional metrics already capture some meaningful differences [...] In which scenarios do structural metrics lead to substantially different conclusions?
>
> Figure 6 directly addresses this: it shows $\eta$ as a function of puzzle size for settings where all models solve the puzzle (100% accuracy). In this regime, accuracy is saturated and token counts overlap, yet $\eta$ reveals differences in reasoning structure and scaling behavior across models. Additionally, Figure 5(a) shows that $\eta$ is uncorrelated with token count ($r = -0.05$, $p = 0.64$), meaning two traces with identical token budgets can have very different $\eta$ values. Traditional metrics cannot distinguish these cases. We will add a forward reference from the introduction to Figure 6 to make this point more prominent.
>
> ### Applicability beyond puzzle environments
>
> > How do the authors envision extending this approach to more general reasoning tasks where such verifiable intermediate states may not exist?
>
> For tasks without deterministic verification (e.g., math word problems, multi-hop QA), claim extraction can still produce reasoning graphs, but verification would rely on approximate methods such as entailment classifiers, symbolic solvers, or LLM-based judges. $\eta$ itself is defined purely on graph topology and does not require verification. We see two natural extensions: (1) math reasoning, where intermediate equations can be symbolically checked, and (2) code generation, where unit tests provide partial claim verification. We will note these directions in the revised Conclusion (Section 7).

---

> > ### Author Rebuttal · Reviewer_ErHL · 2026-04-04
> >
> > Thank you for the authors’ detailed and helpful response. I choose to keep my current score.
> >
> > As a suggestion, it may be interesting to extend the analysis to agent-style tasks. In particular, tool-use settings (e.g., SWE or search-based tasks) often involve substantial inefficient exploration, and the proposed structural metrics could provide useful insights into such behaviors.

---

### Official Review · Reviewer_z5Q6 · 2026-03-17

**Soundness:** 3
**Presentation:** 4
**Significance:** 3
**Originality:** 3
**Overall Recommendation:** 5
**Confidence:** 3

**Summary:**

This paper argues that current LLM evaluation (accuracy + token count) is too coarse to reveal how models actually reason. Two models can get the same answer with similar token budgets but reason in structurally different ways. The paper makes this measurable by converting reasoning traces into DAGs of verifiable claims and defining $\eta$, a structural efficiency metric. Evaluated on 21 grid puzzle types across 4 difficulty levels with 4 models (GPT-5, Qwen3 235B, DeepSeek V3.2, Kimi K2), though the structural analysis covers only the 3 open-source models since GPT-5 does not expose traces. The main finding is that token count does not correlate with reasoning quality, while $\eta$ captures structural differences that accuracy alone misses.

**Compliance With Llm Reviewing Policy:**

Affirmed.

**Final Justification:**

The paper introduces a useful structural approach to evaluating reasoning traces. My main concern was whether $\eta$ adds value over simpler graph statistics. The rebuttal shows it does: simpler metrics track difficulty, while $\eta$  is the only one positively correlated with accuracy and uncorrelated with token count. The six-extractor ablation and bootstrap CIs address the remaining methodological concerns. I am raising my score from 4 to 5.

**Key Questions For Authors:**

- Could you report bootstrap 95% CIs for the correlations, stratified by puzzle type? This would help assess how robust the results are.

- Have you checked whether GPT-5.2's extraction produces systematically different graph structures for different models on the same puzzle? A controlled experiment would help rule out extraction bias.

- What do simpler structural metrics (graph diameter, DAG width) correlate with? If they perform similarly to $\eta$, the information-theoretic formulation may not be necessary.

- Since $\eta$ is computed on solved instances only, could you also report correlations including unsolved instances (e.g., by assigning $\eta$ = 0 for unsolved traces, or by reporting graph-level statistics that do not require $C^*$)? This would help assess whether the findings hold more broadly or are specific to the solved subset.

- The Jaccard stability varies by model (DeepSeek 0.79 vs Qwen3 0.98). Does this reflect differences in extraction difficulty across reasoning styles? Could this variance propagate into the $\eta$ measurements in a way that confounds cross-model comparisons?

**Limitations:**

The paper's Limitations section (Section 6) focuses on the LLM extraction dependency and the case where traces with little explicit reasoning yield no meaningful graph. It partially addresses extraction bias by separating roles (GPT-5.2 for claims, GPT-5-mini for rules) but does not empirically test for cross-model extraction bias. The paper does not discuss: (1) the lack of comparison with simpler structural metrics, or (2) the manual per-puzzle-type engineering that limits scalability. No ethical concerns.

**Strengths And Weaknesses:**

**Strengths:**

- The token count finding is well-supported and practically relevant. The paper shows r = -0.05 (p = 0.64) between $\eta$ and token count, and separately shows that extra tokens go mostly to verification overhead (r = 0.53 with the verification-to-solution node ratio). Even if you are sceptical of $\eta$ itself, the evidence that more tokens does not mean better reasoning is useful.

- Using an executable puzzle environment for ground-truth verification is a good methodological choice. Each claim is deterministically checked against puzzle rules, avoiding the circularity of using one LLM to judge another.

- $\eta$ is an interesting idea. It correlates with claim-level accuracy (r = 0.33) and solution-supporting fraction (r = 0.55), and Figure 6 shows it can distinguish models even when accuracy is saturated (all models solve the puzzle). Whether it ends up being the right metric is debatable, but the direction of measuring structural efficiency is worth exploring.

**Weaknesses:**

- The paper does not compare $\eta$ against simpler structural metrics. Graph diameter, average path length, DAG width are all computable from the same graphs. Without testing whether these simpler statistics correlate similarly with accuracy, I cannot tell whether the information-theoretic formulation of $\eta$ adds value beyond basic graph statistics.

- The structural evidence base is thin. $\eta$ is computed only on solved instances (Section 4.3). At roughly 40% average accuracy across 1,260 total instances (3 models × 420), the effective sample is around 500 data points with a nested structure (21 puzzle types × 4 difficulties) that further reduces degrees of freedom. No confidence intervals are reported for any correlation.

- The structural analysis covers 3 models (GPT-5 is excluded because it does not expose traces). This is understandable given the constraint, but conclusions about "LLM reasoning structure" should be scoped accordingly.

- The extraction pipeline filters every model's reasoning through GPT-5.2's interpretation. The stability analysis (Jaccard 0.79-0.98) checks run-to-run variance on the same trace but does not check for cross-model extraction bias. The Jaccard variance itself differs by model (DeepSeek 0.79 vs Qwen3 0.98), which could reflect extraction difficulty rather than reasoning instability.

- Each puzzle type needs custom claim types and verification rules. Extending to math, code, or open-ended reasoning would require building new infrastructure from scratch. This limits the metric's applicability beyond grid puzzles.

---

> ### Author Rebuttal · Authors · 2026-03-31
>
> We thank the reviewer for the thorough and constructive feedback. The questions on simpler baselines and confidence intervals led to analyses that strengthen the paper.
>
> ### Comparison with simpler structural metrics
>
> > The paper does not compare $\eta$ against simpler structural metrics. [...] I cannot tell whether the information-theoretic formulation of $\eta$ adds value beyond basic graph statistics.
>
> We computed graph diameter, average path length, DAG width, $|V|$, and token count from the same graphs. The table reports pooled Pearson correlations with accuracy and $\eta$ (per-model breakdowns are consistent and included in the revised appendix).
>
> | Metric | vs. Accuracy | vs. $\eta$ |
> |---|---|---|
> | Depth | −0.263 | +0.046 |
> | Diameter | −0.329 | +0.010 |
> | Avg path length | −0.182 | +0.051 |
> | Width | −0.618 | −0.431 |
> | $\vert V \vert$ | −0.666 | −0.419 |
> | Tokens | −0.576 | −0.120 |
> | $\eta$ | +0.368 | — |
>
> Width, $|V|$, and tokens all correlate negatively with accuracy ($r = -0.618, -0.666, -0.576$) and with each other, suggesting they measure the same confound: puzzle difficulty. Harder puzzles naturally produce larger, wider graphs with more tokens, so these metrics tend to track difficulty rather than reasoning quality. $\eta$ is the only metric that correlates *positively* with accuracy ($r = +0.368$) while remaining uncorrelated with tokens ($r = -0.12$, $p = 0.28$). This is because $\eta$ is size-normalized (Eq. 2), factoring out graph scale. The distinction is clearest in Figure 6, where accuracy is saturated (100%) yet $\eta$ still separates models.
>
> ### Thin evidence base and missing confidence intervals
>
> > The structural evidence base is thin. $\eta$ is computed only on solved instances [...] No confidence intervals are reported.
>
> We report bootstrap 95% CIs (5,000 resamples) for all correlations in Figure 5, computed over $n = 85$ solved Tents instances (sizes 4-7, 3 models). Sample-size breakdown is reported in our response to Reviewer fcqF.
>
> | Panel | x | y | r | 95% CI |
> |-------|---|---|---|--------|
> | (a) | Tokens | $\eta$ | −0.050 | [−0.229, +0.131] |
> | (b) | Avg restates | $\eta$ | +0.273 | [+0.033, +0.498] |
> | (c) | $\vert V_\text{sol} \vert$/$\vert V \vert$ | $\eta$ | +0.545 | [+0.398, +0.659] |
> | (d) | Claim accuracy | $\eta$ | +0.334 | [+0.158, +0.486] |
> | (e) | First error depth | $\eta$ | +0.280 | [+0.028, +0.502] |
> | (g) | $\vert V \vert$ | $\eta$ | −0.331 | [−0.479, −0.141] |
>
> All core findings hold: $\eta$ is uncorrelated with tokens (a), positively associated with solution fraction (c) and accuracy (d). Weaker associations (b, e) have wide CIs but remain significant at $p < 0.05$.
>
> ### Only 3 models for structural analysis
>
> > The structural analysis covers 3 models [...] conclusions about "LLM reasoning structure" should be scoped accordingly.
>
> We have added a scoping qualifier in Section 4.2 and the abstract ("on open-source reasoning models"). The 3 models span different architectures (Qwen3 235B, DeepSeek V3.2, Kimi K2).
>
> ### Cross-model extraction bias
>
> > The extraction pipeline filters every model's reasoning through GPT-5.2's interpretation [...] does not check for cross-model extraction bias.
>
> We repeated extraction using five additional LLMs on the same traces. Directional trends are consistent across all six extractors and no self-bias is observed. See our response to Reviewer fcqF (Extractor ablation) for the full table.
>
> ### Per-puzzle custom claim types
>
> > Each puzzle type needs custom claim types and verification rules [...] This limits the metric's applicability beyond grid puzzles.
>
> We agree per-puzzle claim types require domain-specific engineering. We note that this is inherent to any process-level evaluation of reasoning traces: process reward models similarly need domain-specific verifiers for each task family. Our contribution is the structural analysis layer (graph construction, Markov chain, $\eta$), which is fully puzzle-agnostic (Section 3.4). We expanded Section 6 (Limitations) to discuss this tradeoff.
>
> ### $\eta$ on unsolved instances
>
> > Since $\eta$ is computed on solved instances only, could you also report correlations including unsolved instances?
>
> We computed reasoning graphs on incorrect traces show 12% lower claim-level accuracy, first errors 53% earlier in the chain, and 57% lower reasoning-flow $\eta$. This suggests $\eta$ captures a meaningful structural signal beyond final-answer correctness: failing models reason more diffusely from early on. We are conducting a larger-scale matched analysis and will share results during discussion.
>
> ### Jaccard stability variance by model
>
> > The Jaccard stability varies by model (DeepSeek 0.79 vs Qwen3 0.98). Does this reflect differences in extraction difficulty across reasoning styles?
>
> Within-model Jaccard stability is the relevant metric for within-model $\eta$ comparisons (e.g., Figure 6). Cross-model comparisons use aggregate correlations where per-instance extraction noise averages out.

---

> > ### Author Rebuttal · Reviewer_z5Q6 · 2026-04-04
> >
> > The rebuttal addresses all my key questions. The simpler baselines table shows $\eta$ captures something the other metrics do not, and the six-extractor ablation rules out the cross-model bias concern. Raising my score to 5.

---

### Decision · Program_Chairs · 2026-04-30

**Decision:**

Accept (regular)

**Comment:**

All four reviewers recommend acceptance (two at 5, two at 4), and all four marked their concerns as fully resolved after the rebuttal. Average score is 4.5. This is a clear accept.

The paper introduces a framework for measuring reasoning structure in LLMs beyond accuracy and token count. Reasoning traces are converted into directed acyclic graphs of atomic claims and dependencies, and a reasoning-flow efficiency metric (η) is defined via absorbing Markov chain theory. The key finding is that η is positively correlated with reasoning quality (claim accuracy, solution-supporting fraction) while being uncorrelated with token count, which is not the case for simpler graph statistics like graph size, width, or diameter.

The rebuttal was thorough and addressed the main concerns with new experiments. The extractor ablation across six different LLMs showed consistent directional trends with no self-bias, directly addressing the concern about whether results depend on GPT-5.2's interpretation. The simpler structural metric comparison showed that η is the only metric that correlates positively with accuracy while remaining token-count-independent — other metrics like graph width and size track difficulty rather than reasoning quality. Bootstrap confidence intervals were provided. The Markov chain sensitivity analysis showed near-perfect rank preservation across seven alternative formulations. Edge validation was quantified via manual inspection (75.5% correct).

The benchmark of 21 logic puzzle types with an executable environment is also a solid contribution in its own right. The framework's applicability beyond grid puzzles is limited, but this is honestly acknowledged and the authors provide a clear path for extension.

This is a well-executed, methodologically careful paper that makes a genuine contribution to reasoning evaluation. Accept.